# AdaProx: A Novel Method for Bilevel Optimization under Pessimistic Framework

Ziwei Guan[1,2]* Daouda Sow[1], Sen Lin[1,3], Yingbin Liang[1]
[1]Ohio State University, [2]Meta Platform Inc., [3]University of Houston
guanziwei@meta.com, sow.53@osu.edu, slin50@central.uh.edu, liang.889@osu.edu

As a powerful framework for various machine learning problems, bilevel optimization has attracted significant attention recently. While many modern gradient-based algorithms have been devised for optimistic bilevel optimization (OBO), pessimistic bilevel optimization (PBO) is much less explored and there is almost no formally designed algorithms for nonlinear PBO with provable convergence guarantee. To fill this gap, we investigate PBO with nonlinear inner- and outer-level objective functions in this work. By leveraging an existing reformulation of PBO into a single-level constrained optimization problem, we propose an Adaptive Proximal (AdaProx) method which features novel designs of adaptive constraint relaxation and accuracy level in order to guarantee an efficient and provable convergence. We further show that AdaProx converges sublinearly to an $\epsilon$-KKT point, and characterize the corresponding computational complexity. Our experiments on an illustrative example and the robust hyper-representation learning problem validate our algorithmic design and theoretical analysis. To the best of our knowledge, this is the first work that develops principled gradient-based algorithms and characterizes the convergence rate for PBO under nonlinear settings.

## 1. Introduction

Originated from the economic and operation research studies [1, 2], bilevel optimization has attracted extensive attention recently in the machine learning community. Many machine learning problems can be naturally captured by the bilevel optimization framework such as meta-learning [3, 4], reinforcement learning [5, 6], network architecture searching [7], etc. Bilevel optimization typically takes the following form

$$\text{(OBO problem)} \quad \min_{x \in \mathcal{X}} \min_{y \in \mathcal{S}(x)} f(x, y), \qquad \text{where} \ \ \mathcal{S}(x) = \arg\min_{y \in \mathbb{R}^m} g(x, y), \qquad (1)$$

where $f(x, y)$ and $g(x, y)$ are the outer- and inner-level objective functions, respectively, and the support set $\mathcal{X} \subseteq \mathbb{R}^p$ is typically convex. For a fixed $x \in \mathcal{X}$, the inner optimization finds a set $\mathcal{S}(x)$ that collects all points $y$ that minimize the inner function $g(x, \cdot)$. Then, the outer-level function $f(x, y)$ is minimized over $y$ in the set $\mathcal{S}(x)$ jointly with $x \in \mathcal{X}$. The above problem is referred to as **optimistic bilevel optimization (OBO)**, because the outer-level minimizes over $y \in \mathcal{S}(x)$, which allows the minimization over $x$ to be over a beneficial loss value. Such OBO problems have been extensively studied in the past, e.g., Harker and Pang [8], Outrata [9], Lignola and Morgan [10], Dempe et al. [11] and Sinha et al. [12], Liu et al. [13]. More recently, many studies have developed various fast and scalable algorithms and provided the convergence rate guarantee for these algorithms [14–21]. Readers can refer to Section 1.2 for more detailed discussion of the related work.

As an equally important class of bilevel problems, **pessimistic bilevel optimization (PBO)** takes the following formulation

$$\text{(PBO probelm)} \quad \min_{x \in \mathcal{X}} \max_{y \in \mathcal{S}(x)} f(x, y), \qquad \text{where} \ \ \mathcal{S}(x) := \arg\min_{y \in \mathbb{R}^m} g(x, y). \qquad (2)$$

For each given $x$, the inner optimization also collects all minima of the inner function $g(x, \cdot)$ into a set-value function $\mathcal{S}(x)$. Then the outer-level function $f(x, y)$ is first maximized over $y \in \mathcal{S}(x)$, and

---

*Ziwei completed his contribution to this work at OSU

Second Conference on Parsimony and Learning (CPAL 2025).

then minimized over the outer variable $x$. Intuitively, the maximization finds the worst case of the outer-level function over $y \in \mathcal{S}(x)$, and is hence called the **pessimistic** problem.

PBO can capture many real-world machine learning applications. For example, consider a robust hyperparameter learning problem where we seek to learn the best hyperparameters that are robust to the model learning. Specifically, given a hyperparameter $x \in \mathcal{X}$, the inner problem aims to find the optimal model on the training datasets for a training loss function $g(x, y)$ as $\min_{y \in \mathbb{R}^m} g(x, y)$, where multiple optimal $y$ may exist and are collected into a set $\mathcal{S}(x)$. However, due to the randomness of the training dataset and the algorithm design, the validation loss of the learned model on a different validation dataset could be as large as $\max_{y \in \mathcal{S}(x)} f(x, y)$. To guarantee a more robust learning performance, we aim to learn the hyperparameter $x$ that excels in the worst case as $\min_{x \in \mathcal{X}} \max_{y \in \mathcal{S}(x)} f(x, y)$. Clearly, such a robust hyperparameter learning problem falls under the PBO framework. Another PBO example of the hyper-representation problem is in Section 5.2.

In contrast to OBO, PBO is more challenging to solve due to its min-max nature in the outer optimization, and still remains much less studied in the literature. Particularly, the previous studies of PBO mainly focused on the existence of optimal solutions and the characterization of optimality condition [22–24], which have shown that the optimality condition of PBO is more strict than that of OBO. In particular, [22, 23] reformulated PBO into constrained optimization via the KKT conditions, which facilitates the characterization of the optimality conditions. Besides, the design of algorithms therein was mainly restricted to *linear* bilevel optimization [25–27] and lacked convergence guarantees. For more general PBO problems, a recent work [28] proposed a Scholtes relaxation scheme for PBO and proved its convergence asymptotically. To the best of our knowledge, no prior work developed the *non-asymptotic* convergence rate for PBO for the general function class.

In fact, although the reformulated PBO problem as in [22, 23] falls into the general framework of constrained optimization, several challenges still need to be addressed in order to design efficient algorithms with provable convergence guarantees. (a) Studies on nonconvex constrained optimization such as [29, 30] typically make assumptions of uniform Slater condition and strong feasibility, which are not satisfied by reformulated PBO problems in general. (b) Reformulation typically introduces bias errors for estimating gradients, which are not present for standard constrained optimization. Such bias errors can significantly affect the convergence of gradient-based algorithms. (c) Reformulation typically introduces relaxation to smooth the objective function and facilitate easy implementation of gradient-based methods. The relaxation parameters need to be selected in a principled way to guarantee the equivalence of the reformulation to the original PBO problem.

In this paper, we address the aforementioned challenges and develop the first-known principled gradient-based algorithms for PBO that enjoy convergence guarantees and are easy to implement.

## 1.1. Main Contributions

We summarize our main contributions as follows.

**Algorithmic design.** We propose a novel Adaptive Proximal (AdaProx) method for PBO problems, which is the first-known provably convergent first-order algorithm for PBO. Although AdaProx takes standard constrained convex optimization solvers such as switching gradient (SG) [29, 31, 32] and primal-dual (PD) [30, 33] as subroutines, it further features the following new designs: (a) a novel relaxation on the constraint that is *adaptive* to the iteration $k$ in order to guarantee the Slater condition and strong feasibility for the constraints; and (b) simple yet efficient estimators to approximate function values and gradients of the constraints to control the bias errors.

**Convergence rate analysis.** We first establish the connection between the value functions of the original PBO and that of the reformulated problem, which shows that the reformulation introduces controllable deviations from the original PBO. We then show that AdaProx converges to an $\epsilon$-KKT point of the reformulated problem with a sublinear rate, where the KKT condition serves as a necessary condition for the local optimality.

Technically, beyond the standard analysis for constrained optimization, we need to devise a few new techniques to deal with the specific challenges here due to the nature of bilevel optimization. (a) We need to characterize the impact of the adaptive relaxation of the constraints on the convergence error of the proximal point iterations. (b) Our analysis needs to upper-bound the bias error in the gradient estimation due to the inner-level problem and control such a bias error to a desirable accuracy level.

**Numerical verification.** We evaluate the numerical performance of AdaProx that takes SG and PD as subproblem solvers, which we respectively refer to as AdaProx-SG and AdaProx-PD. Our experimental results show that AdaProx can converge to the global optima of the studied problems with fast rate, which validate our algorithmic design and theoretical analysis. Further, compared to AdaProx-PD, AdaProx-SG has a better track of the constraint violation and, as a tradeoff, the convergence of its outer-level objective appears to be less stable.

## 1.2. Related Works

**Pessimistic Bilevel Optimization:** On the theoretical side, previous studies focused on identifying the existence of solution [34–36], and characterizing the conditions of optimality [22, 37, 38]. Different reformulations have been suggested to make PBO more tractable, such as changing PBO to constrained optimization via the KKT conditions [22, 23], incorporating the inner-loop problem into the outer-loop problem as an additional penalty term [36, 39] and expressing pessimism in the form of two-player game at the inner-level [36]. From a numerical perspective, algorithms were only designed under restrictive settings such as linear PBO [27] and quadratic-linear PBO [40]. For the general PBO, Wiesemann et al. [25] proposed a finite-dimensional approximation method, which restricted the support of inner-level problems to be a finite subset of $\mathbb{R}^n$, i.e., $Y_k \subseteq \mathbb{R}^n$ and $|Y_k| \leq \infty$, and enlarged the cardinality of $Y_k$ to approximate the original problem gradually. In contrast to the above studies, this paper provides a novel proximal method for general PBO functions and provides the first-known convergence analysis. Zeng [26] studied the general PBO problem and gave a tight relaxation which has the same global solution of the original PBO and could be reduced to OBO in specific settings, including linear PBO, mixed-integer PBO, and coupled pessimistic constrained PBO. A recent work [28] proposed a Scholtes relaxation scheme for PBO with inner-level problem having a functional constraint and showed that the stationary points of a sequence of relaxed problems converge to the stationary point of the original PBO problem. We further refer the readers to the survey work [12, 24], which provided a comprehensive summary of the literature on PBO.

**Recent Advances in OBO:** The gradient-based algorithms have become popular for solving the bilevel optimization problem with unique inner-minimum, due to their simplicity and scalability. For example, to compute the gradient of the outer-level optimization efficiently, both approximated implicit differentiation (AID) [41–43] and iterative differentiation (ITD) approaches [41, 44, 45] have been widely studied. Asymptotic convergence analysis was studied in, e.g., Franceschi et al. [3], Shaban et al. [46], and recently Ji et al. [4], Ji and Liang [43], Grazzi et al. [47], Ji et al. [48] provided the non-asymptotic convergence rate analysis. Another line of studies [14, 16, 49] utilized the gradient sequential averaging method to solve the optimistic bilevel optimization with single inner-optimum. More recently, there has been substantial interest in OBO problems with multiple inner minimal points. Specifically, a recent work [15] proposed a gradient-based and hessian-free algorithm for solving such OBO problems, and provided the non-asymptotic analysis therein. The work [50] provided a dynamic barrier gradient method. Later, the work [51] proposed a new convergence metric for the case where inner problem does not have the strongly convex assumption, and then designed a zeroth-order method for the suggested metric. The work [52] developed a new convergent method with the inner-level problem being constrained optimization. The PBO problem we consider here is more challenging than OBO, due to the minimax nature in the outer problem.

**Generic Nonconvex Constrained Optimization:** The convex constrained optimization problem has been extensively studied in the literature [53–57]. The constrained optimization with nonconvex functional constraints has recently attracted increasing attention. Several algorithms have been proposed and shown to converge efficiently, including proximal method [29, 30, 58], sequentially

quadratic programming [59], and augmented primal-dual method [60]. In this paper, although we adopt an approach that formulates PBO into constrained optimization with nonconvex objective and constraint functions, several challenges arise due to the special structure of PBO. Our contributions here lie in new algorithm design components as well as the convergence analysis that handles those new design components.

## 2. Problem Formulation

We study the PBO problem in eq. (2) in this paper. We assume that the constraint set $\mathcal{X}$ is convex and closed set. Usually $\mathcal{X}$ has a simple structure, e.g., simplex or closed interval, and the orthogonal projections onto $\mathcal{X}$ is easy to compute. We make the necessary assumptions on $f$ and $g$ as follow:

**Assumption 1.** For any given $x \in \mathcal{X}$, $f(x, y)$ is a concave function on $y$, and $g(x, y)$ is a convex function on $y$. Let $\theta = (x, y)$ and $\theta' = (x', y')$. $f(x, y)$ and $g(x, y)$ are twice continuously differentiable with Lipschitz continuous gradient and Hessian, i.e., there exist constants $L_f$, $L_g$, $\rho_f$ and $\rho_g$, such that for any $x, x' \in \mathcal{X}$, $y, y' \in \mathbb{R}^m$, we have

$$\|\nabla f(\theta) - \nabla f(\theta')\|_2 \le L_f \|\theta - \theta'\|_2, \qquad \|\nabla g(\theta) - \nabla g(\theta')\|_2 \le L_g \|\theta - \theta'\|_2,$$
$$\|\nabla^2 f(\theta) - \nabla^2 f(\theta')\|_F \le \rho_f \|\theta - \theta'\|_2, \qquad \|\nabla^2 g(\theta) - \nabla^2 g(\theta')\|_F \le \rho_g \|\theta - \theta'\|_2,$$

where $\nabla h$ and $\nabla^2 h$ denote the gradient and the Hessian matrix of a function $h$ with respect to (w.r.t.) $\theta$, respectively, and $\| \cdot \|_F$ denotes the Frobenius norm of matrices. Moreover, for all $x \in \mathcal{X}$ and $y \in \mathbb{R}^m$, there exists a constant $\kappa > 0$ such that $\lambda_{min}(\nabla^2_{yy} g(x, y)) > \kappa$ for all $\nabla_y g(x, y) \ne 0$, where $\lambda_{min}(\cdot)$ denotes the minimum eigenvalue of a matrix.

### 2.1. Single-level Reformulation

In this section, we introduce the reformulation of PBO in eq. (2) to a constrained optimization problem [22, 23] (see also the survey work [12, 24]).

In order to solve the PBO problem in eq. (2), let $g^*(x) := \min_{y \in \mathbb{R}^m} g(x, y)$ and replace the set $\mathcal{S}(x)$ by its equivalent form $\mathcal{S}(x) = \{y \in \mathbb{R}^m : g(x, y) - g^*(x) \le 0\}$. In this way, PBO problem can be reformulated equivalently as:

$$\min_{x \in \mathcal{X}} \max_{y \in \mathbb{R}^m} \quad f(x, y), \quad \text{s.t.} \quad g(x, y) - g^*(x) \le 0. \tag{3}$$

In order to solve eq. (3) efficiently, we introduce constraint relaxation. For any small positive constants $\alpha$ and $\xi$, an $(\alpha, \xi)$-relaxation of the problem in eq. (3) is typically introduced as follows [15, 61]

$$\min_{x \in \mathcal{X}} \max_{y \in \mathbb{R}^m} \quad f(x, y), \quad \text{s.t.} \quad g(x, y) - g^*_\alpha(x) - \xi \le 0, \tag{4}$$

where $g^*_\alpha(x) := \min_{y \in \mathbb{R}^m} g_\alpha(x, y) := g(x, y) + \frac{\alpha}{2}\|y\|_2^2$. The $\ell_2$-regularization ensures $g_\alpha(x, y)$ to be strongly convex on $y$, and hence the solution $y^*_\alpha(x) := \arg\min_{y \in \mathbb{R}^m} g_\alpha(x, y)$ is unique for any given $x \in \mathcal{X}$. The regularization also ensures that $g^*_\alpha(x)$ is differentiable, and its gradient takes the form of $\nabla_x g^*_\alpha(x) = (\nabla_x g_\alpha(x, y))|_{y=y^*_\alpha(x)}$. Besides, the positive constant $\xi$ in the constraint guarantees that the relaxed problem eq. (4) has at least one strictly feasible point for any given $x \in \mathcal{X}$, which is vital for solving the problem efficiently.

The "max" over $y$ in eq. (4) can be further removed via the KKT conditions which serve as the constraints that the optimal $y$ should satisfy. This simplifies the min-max problem in eq. (4) to an equivalent single-level constrained minimization problem as follows [12, 62].

$$\min_{x \in \mathcal{X}, y \in \mathbb{R}^m, w \in \mathbb{R}_+} \quad f(x, y) \quad \text{s.t.} \quad g(x, y) - g^*_\alpha(x) - \xi \le 0$$
$$- \nabla_y f(x, y) + w \nabla_y g(x, y) = 0,$$
$$w(g(x, y) - g^*_\alpha(x) - \xi) = 0, \tag{5}$$

where $w$ is the slackness variable introduced by the KKT-conditions. Compared to eq. (4), we have two additional inequality constraints in eq. (5) corresponding to the KKT conditions for $y$ attaining

the maximum of $f(x, y)$ given $g(x, y) - g_\alpha^*(x) - \xi \leq 0$. To further simplify the notation, let $z = (x, y, w)$ and $\mathcal{Z} = \mathcal{X} \times \mathcal{Y} \times \mathcal{W}$. Here, we require $y$ and $w$ to belong to bounded sets $\mathcal{Y}$ and $\mathcal{W}$ for the ease of the algorithm design later on. We further change each equality constraint in eq. (5) into two equivalent inequality constraints, and then obtain

$$\min_{z \in \mathcal{Z}} \ f(z) \qquad \text{s.t.} \ \ h(z) := \begin{pmatrix} g(x, y) - g_\alpha^*(x) - \xi \\ -\nabla_y f(x, y) + w \nabla_y g(x, y) \\ \nabla_y f(x, y) - w \nabla_y g(x, y) \\ w(g(x, y) - g_\alpha^*(x) - \xi) \\ -w(g(x, y) - g_\alpha^*(x) - \xi) \end{pmatrix} \leq 0. \qquad (6)$$

Although the reformulation in eq. (6) of original PBO takes several relaxations, we will show in Section 4.1 that their change of the problem can be made as small as possible by choosing the relevant parameters properly. Hence, in this paper, we will develop an algorithm to solve the reformulated problem in eq. (6), which will solve the original PBO in eq. (2) to any desired target accuracy.

## 3. Adaptive Proximal Method

In this section, our aim is to solve the problem in eq. (6). Since the objective and constraint functions are both possibly nonconvex, we propose a novel adaptive proximal point method called AdaProx (see Algorithm 1). Due to the specific structure that PBO problems have, our method differentiates from the generic method for solving nonconvex optimization with nonconvex constraints [29, 30] in several aspects as we elaborate below.

---

**Algorithm 1** Adaptive Proximal (AdaProx) Method

---

1: **Input:** Number of iterations $K$, $T$, relaxation level $\beta$, regularization parameter $\sigma$, and initial point $\tilde{z}_1$.
2: **for** $k = 1, ..., K$ **do**
3:    Set the $k$th subproblem ($\mathrm{P}_k$) as in eq. (7)
4:    Call a standard solver such as SG and PD (see appendix A) to solve $\mathrm{P}_k$ to a $\frac{\beta}{2K}$-accurate solution
5: **end for**
6: Pick $\hat{k}$ from $\{1, \ldots, K\}$ uniformly at random
7: **Output:** $\tilde{z}_{\hat{k}}$

---

At each iteration $k$, we construct a subproblem $\mathrm{P}_k$ from eq. (6) by adding regularizers centered at the current solution ($\tilde{z}_k$) in both the objective and constrained functions as follows:

$$\min_{z \in \mathcal{Z}} \ f_k(z) := f(z) + \frac{\sigma}{2}\|z - \tilde{z}_k\|_2^2 \qquad \text{s.t.} \ \ h_k(z) := h(z) + \frac{\sigma}{2}\|z - \tilde{z}_k\|_2^2 - \frac{k\beta}{K} \leq 0, \qquad (7)$$

By setting the $\sigma$ large enough, both the objective and the constrained functions are strongly convex.

**Challenge and novel designs:** Note that the proximal method for generic constrained nonconvex problems [29, 30] made assumptions of uniform Slater condition and strong feasibility for the constraints. However, the constraints in eq. (6) do not satisfy these conditions. The inequality constraints corresponding to the KKT conditions cannot be strictly satisfied simultaneously because they are exactly opposite to each other (e.g., the second and third terms, and the fourth and fifth terms in eq. (6)). To address this, we introduce two novel ingredients in our design of the algorithm.

- **Adaptive constraint relaxation**: We devise a relaxation term of $-\frac{k\beta}{K}$ in the constraints in eq. (7) that is adaptive to the subproblem index $k$. By gradually increasing such a relaxation by $\frac{\beta}{K}$ in each iteration, $\tilde{z}_{k+1}$ (as the solution of $P_k$) is still $\frac{\beta}{2K}$ strictly feasible for constraints in the next subproblem $P_{k+1}$, even if it may violate the current constraints by $\frac{\beta}{2K}$. This design guarantees that each subproblem $\mathrm{P}_k$ has a strict feasible point.

- **Accuracy level design:** To apply a standard solver for constrained convex optimization (line 4 in Algorithm 1) to solve $\mathrm{P}_k$, we design a specific accuracy level of $\frac{\beta}{2K}$, and obtain a solution of $\tilde{z}_{k+1}$, which will serve as the center point of the regularizers for the next subproblem

$P_{k+1}$. Such an accuracy level of $\frac{\beta}{2K}$ ensures that $\tilde{z}_{k+1}$ can violate the constraints of $P_k$ by no more than $\frac{\beta}{2K}$, which together with the adaptive constraint relaxation guarantees that the subproblems are solved with provable error controls.

After $K$ iterations, the algorithm picks one of the $\tilde{z}_k$ uniformly at random as the output.

# 4. Theoretical Results

## 4.1. Connection to Original PBO

In the reformulation of PBO in Section 2.1, several relaxation steps were taken including the $\ell_2$-regularization and constraint relaxation in eq. (4), the bounded set $\mathcal{W}$ for the variable $w$ in eq. (6) and the bounded set $\mathcal{Y}$. We require that $\mathcal{Y}$ is large enough to include all feasible points of the relaxed problem in eq. (4).

**Assumption 2.** For all $x \in \mathcal{X}$, $\mathcal{S}_{\alpha,\xi}(x) \subseteq \mathcal{Y}$, with $\mathcal{S}_{\alpha,\xi}(x) := \{y \in \mathbb{R}^m : g(x,y) - g_\alpha^*(x) - \xi \leq 0\}$

In the following result, we show that the change of the problem due to those relaxations can be made as small as possible by choosing the relevant parameters properly.

**Proposition 1.** *Suppose Assumption 1 holds. For any fixed $x \in \mathcal{X}$, define the value function for the original problem in eq. (2) as $\Phi(x) := \max_{y \in \mathbb{R}^m}\{f(x,y) : y \in \mathcal{S}(x)\}$, Moreover, let the value function for our reformulated problem in eq. (6) as $\Phi_{\alpha,\xi}(x) = \max_{y \in \mathcal{Y}, z \in \mathcal{W}}\{f(x,y) : h(x,y,z) \leq 0\}$. We set $\mathcal{W} := \{w : 0 \leq w \leq \frac{\Delta_f}{\xi}\}$ with $\Delta_f = \max_{x,x' \in \mathcal{X}, y,y' \in \mathcal{Y}} |f(x,y) - f(x',y')|$. Then for every $x \in \mathcal{X}$, we have*

$$|\Phi(x) - \Phi_{\alpha,\xi}(x)| \leq \mathcal{O}(\sqrt{\alpha}) + \mathcal{O}(\sqrt{\xi}).$$

Proposition 1 indicates that the solution to eq. (6) can be arbitrarily close to that of the original PBO problem. Thus, solving eq. (6) will provide a desirable solution to the PBO problem in eq. (2). The proof of Proposition 1 is provided in Appendix C.2.

## 4.2. Convergence of Solvers for Subproblems

Since the convergence of AdaProx depends on the solvers that we adopt for solving the subproblems, in this subsection we analyze the convergence of the two popular solvers SG and PD as described in Appendix A.

**Technical challenge:** Compared to the standard analysis for constrained optimization [29, 30] which has exact access of the function value and gradient oracles, our analysis here needs to carefully deal with the bias error of the function estimation $\hat{h}_k(z_t; \hat{y}_N^t)$ and the bias error of the Jocobian matrix estimation $\widehat{\nabla} h_k(z_t; \hat{y}_N^t)$. This is because $\hat{y}_N^t$ is only an approximation of a minimum point of the inner function of PBO. Furthermore, the Lipschitz smoothness of both objective and constraint functions in PBO need to be established by exploiting the bilevel problem structure.

**Proposition 2.** *Suppose Assumption 1 holds. Each entry of $h(z)$ in eq. (6) is $L_c$-gradient Lipschitz for some constant $L_c > 0$.*

The above proposition ensures that if we let $\sigma = \max\{2L_f, 2L_c\}$, both the objective and constrained functions of the subproblems in eq. (7) in AdaProx are $\frac{\sigma}{2}$-strongly convex function, for which we introduce the following criterion to characterize its convergence.

**Definition 1.** *Let $z_k^*$ be the solution to the constrained optimization in eq. (7) and $\epsilon \geq 0$ be a constant. We say that $z \in \mathcal{Z}$ is an $\epsilon$-accurate solution if $f_k(z) \leq f_k(z^*) + \epsilon$ and $h_k(z) \leq \epsilon$.*

We characterize the convergence performance of the SG and PD solvers (see Algorithms 2 and 3 in appendix A) used for solving the subproblems in eq. (7) in AdaProx in the following two theorems.

**Theorem 1.** *Suppose that Assumption 1 holds. Let $\sigma = \max\{2L_f, 2L_c\}$. And set the parameters $\gamma_t = \mathcal{O}\left(\frac{1}{t}\right)$, $T = \mathcal{O}\left(\frac{1}{\epsilon}\right)$ and $N = \mathcal{O}\left(\log\left(\frac{1}{\epsilon}\right)\right)$. Then the output $\tilde{z}_{k+1}$ of SG (i.e., Algorithm 2 in appendix A) is*

|  | first-order orcale | second-order orcale |
|---|---|---|
| SG | $\mathcal{O}\left(\frac{1}{\epsilon}\log\left(\frac{1}{\epsilon}\right)\right)$ | $\mathcal{O}\left(\frac{1}{m\epsilon}\right)$ |
| PD | $\mathcal{O}\left(\frac{1}{\sqrt{\epsilon}}\log\left(\frac{1}{\epsilon}\right)\right)$ | $\mathcal{O}\left(\frac{1}{\sqrt{\epsilon}}\right)$ |

Table 1: Comparison between SG and PD solvers on the first- and second-order oracle computation

$\epsilon$-accurate for solving the subproblem $P_k$ in eq. (7) in AdaProx, which satisfies $f_k(\tilde{z}_{k+1}) - f_k(z_k^*) \leq \epsilon$, and $\max_j\{(h_k(\tilde{z}_{k+1}))_j\} \leq \epsilon$.

Theorem 1 shows that SG can solve the $k$th subproblem in eq. (7) to any arbitrary accuracy level $\epsilon$ with a gradient computation complexity of $TN = \mathcal{O}\left(\frac{1}{\epsilon}\log(\frac{1}{\epsilon})\right)$. Furthermore, the computational complexity of the second order Jacobian matrix is upper-bounded by $T/(2m+3) = \mathcal{O}(\frac{1}{m\epsilon})$, since at each iteration SG at most computes one row of the matrix in line 10 of Algorithm 2.

**Theorem 2.** *Suppose that Assumption 1 holds. Let $\sigma = \max\{2L_f, 2L_c\}$. And set parameters $\gamma_t = \mathcal{O}(t)$, $\eta_t = \mathcal{O}(t)$, $\tau_t = \mathcal{O}\left(\frac{1}{t}\right)$, $\theta_t = \frac{\gamma_{t+1}}{\gamma_t}$, $T = \mathcal{O}\left(\frac{1}{\sqrt{\epsilon}}\right)$, and $N = \mathcal{O}\left(\log\left(\frac{1}{\epsilon}\right)\right)$. Then the output $\tilde{z}_{k+1}$ of PD (Algorithm 3 in Appendix A) is $\epsilon$-accurate for solving the subproblem $P_k$ in eq. (7) in AdaProx, which satisfies $f_k(\tilde{z}_{k+1}) - f_k(z_k^*) \leq \epsilon$, and $\max_j\{(h_k(\tilde{z}_{k+1}))_j\} \leq \epsilon$, which indicates that $\tilde{z}_{k+1}$ is an $\epsilon$-accurate solution of the $k$th-subproblem in eq. (7).*

Theorem 2 shows that PD can solve the $k$th subproblem in eq. (7) to any prescribed $\epsilon$ with a gradient computation complexity of $TN = \mathcal{O}\left(\frac{1}{\sqrt{\epsilon}}\log\left(\frac{1}{\epsilon}\right)\right)$. Moreover, since PD needs the information of the entire Jacobian matrix at line 5 of Algorithm 3 (i.e., eq. (14)), the computation complexity of its second order oracle equals $T = \mathcal{O}\left(\frac{1}{\sqrt{\epsilon}}\right)$.

We provide the comparison of SG and PD in Table 1. It can be seen that PD has a lower complexity on the first-order oracle compared to SG. Their complexity comparison of the second-order computation depends on the dimension $m$ and the accuracy level $\epsilon$. If $m\sqrt{\epsilon} > 1$, SG has a lower complexity; and otherwise PD outperforms SG.

## 4.3. Analysis of AdaProx

Since the problem in eq. (6) generally has a nonconvex objective function and nonconvex constraints, we aim to provide the convergence guarantee for AdaProx to an $\epsilon$-KKT point [29, 30] as below.

**Definition 2.** Consider the constrained optimization problem in eq. (6). Let $q$ be the dimension of $h(z)$ and $\mathcal{N}(z;\mathcal{Z})$ be the normal cone to $\mathcal{Z}$ at $z$. Denote $\text{dist}(z,\mathcal{N}) := \min_{z'\in\mathcal{N}}\{\|z-z'\|_2\}$. A point $\hat{z} \in \mathcal{Z}$ is an $\epsilon$-KKT point if and only if there exist $z \in \mathcal{Z}$ and $\lambda \in \mathbb{R}_+^q$, such that $h(z) \leq \epsilon$, $\|z-\hat{z}\|_2^2 \leq \epsilon$, $\sum_{i=1}^q |\lambda_i h_i(z)| \leq \epsilon$, and $\text{dist}\left(\nabla f(z) + \langle\nabla h(z),\lambda\rangle, -\mathcal{N}(z;\mathcal{Z})\right)^2 \leq \epsilon$. Further, a random $\hat{z} \in \mathcal{Z}$ is a stochastic $\epsilon$-KKT point if there exist $z \in \mathcal{Z}$ and $\lambda \geq 0$ such that the same requirements of $\epsilon$-KKT hold in expectation.

The KKT condition is the necessary condition for local optimality [63, 64] for constrained optimization. Here, we will show that AdaProx in Algorithm 1 converges to an $\epsilon$-KKT point in expectation taken over the randomness of the algorithm (the random generation of index $\hat{k}$) for constrained nonconvex optimization problems. Before the analysis, we make the following boundedness assumption on the optimal dual variable, which is standard in the literature [30, 65].

**Assumption 3.** For each subproblem $P_k$, the optimal dual variable $\lambda_k^*$ is uniformly bounded, i.e., there exists a constant $B \geq 0$ such that $\|\lambda_k^*\|_1 \leq B$ holds for all $k = 1, \ldots, K$.

**Theorem 3.** *Suppose Assumptions 1 and 3 holds. Given $\tilde{z}_1$ that is $\frac{\beta}{2K}$ strictly feasible of $(P_1)$. Let $\sigma = 2\max\{L_f, L_c\}$, where $L_c$ is determined in Proposition 2. Set $K = \mathcal{O}(\frac{1}{\epsilon})$ and $\beta = \mathcal{O}(\epsilon^2)$. Then we have $\tilde{z}_{\hat{k}}$ is an $\epsilon$-KKT point of eq. (6) in expectation that takes over randomness of $\hat{k}$.*

Theorem 3 shows that Algorithm 1 is guaranteed to solve problem in eq. (6) to arbitrary accuracy $\epsilon$ with $\mathcal{O}(\frac{1}{\epsilon})$ calls of the subproblem solver. Since all the requirements of theorems 1 and 2 hold, the

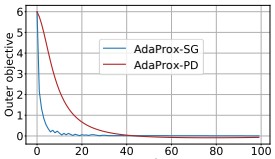
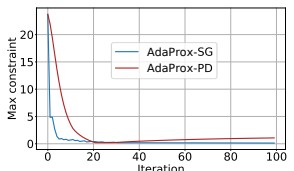

| Algo | $m = 512$ | $m = 1024$ |
|---|---|---|
| AdaProx-SG | 0.29 | 0.57 |
| AdaProx-PD | 0.32 | 0.65 |

Figure 1: Comparison of AdaProx-PD and AdaProx-SG for the illustrative example in eq. (8)

Table 2: Iteration time in (s), running time of each iteration of AdaProx-SG and AdaProx-PD scales similarly on dimension $m$ but with AdaProx-SG slightly faster.

first- and second-order oracle complexity immediately follows by applying those theorems. Compared with results in standard constrained optimization [29, 30], since our algorithm here features a novel adaptive relaxation on the constraint, we need to develop new analysis to characterize the impact of such adaptive relaxation on convergence error of the proximal point iterations.

# 5. Experiments

In this section, we conduct experimental studies on two specific problems to verify that the proposed AdaProx with SG and PD as subproblem solvers achieves desirable performance. Since there was not any well developed algorithms in the literature for general PBO, our focus here is on whether AdaProx returns an optimal solution and how the subsolvers of SG and PD compare with each other in their performance.

## 5.1. Illustrative Example

Consider the following example:

$$\min_{x \in \mathbb{R}} \max_{y \in \mathcal{S}(x)} -xy \quad s.t. \quad x^2 + y^2 - 1 \leq 0, \tag{8}$$

where $\mathcal{S}(x)$ is the set of solutions to the following inner-level optimization with a fixed $x \in \mathbb{R}$, $\min_{y \in \mathbb{R}} g(x, y)$, with $g(x, y) = |y - |x||^3$, when $|y| \geq |x|$, otherwise, $g(x, y) = 0$. It is clear that $\mathcal{S}(x) = \{y \in \mathbb{R} : |y| \leq |x|\}$ and $g^*(x) = 0$. For any fixed $\alpha$, $g^*_\alpha(x) = 0$. More details about the KKT reformulation and the exact forms of gradients could be found in Appendix H.

Figure 1 shows the performance of both AdaProx-SG (Algorithms 1 and 2) and AdaProx-PD (Algorithms 1 and 3) in solving the problem eq. (8), where the x-axis denotes the iteration number. It is clear that both algorithms solve the objective function to its global minimum efficiently. Besides, as illustrated in the left figure in Figure 1, AdaProx-SG converges at a faster rate than AdaProx-PD. This is because AdaProx-SG enforces the constraints only when the threshold $\epsilon$ is violated and will focus solely in minimizing the outer objective $f$ when all the constraints are less than $\epsilon$. Whereas AdaProx-PD will always minimize the Lagrangian, which may result in unnecessary delays in minimizing $f$ when all the constraints are satisfied. Moreover, the left figure of Figure 1 indicates that the constraint violation in AdaProx-SG decreases much faster than that in AdaProx-PD. Recall that the update direction of AdaProx-PD is $\nabla f_k(z_t) + \langle \nabla h_k(z_t), \lambda_{t+1} \rangle$, where the $i$-th constraint gets penalized when the $i$-th entry of $\lambda$ is large enough. Since AdaProx-PD updates the primal variables based on the constraints' value after observing the updates of $\lambda$, it is not hard to tell that the decrease of constraint violation would be slow if the stepsize for updating $\lambda$ is small.

## 5.2. Learning Robust Hyper-representation

In the hyper-representation (HR) [47, 66] problem, the goal is to find good representations of the data that can be used for subsequent regression/classification problem by following a two-phase optimization process. The PBO framework can be used to robustly learn such representations. More specifically, we consider the following formulation:

$$\min_{\Lambda \in \mathbb{R}^{d \times m}} \max_{w^* \in \mathcal{S}_\Lambda} \mathcal{L}\left(h_\Lambda(X_1)w^*, Y_1\right) \qquad \text{with} \quad \mathcal{S}_\Lambda = \underset{w \in \mathbb{R}^m}{\operatorname{argmin}} \mathcal{L}(h_\Lambda(X_2)w, Y_2) \tag{9}$$

where $h_\Lambda(\cdot)$ is the embedding model (linear transformation in this case) parameterized by the matrix $\Lambda$, and the vector $w$ corresponds to the parameters of a linear regression/classification model. $X_1 \in \mathbb{R}^{n_1 \times d}$ and $X_2 \in \mathbb{R}^{n_2 \times d}$ are the matrices of outer (validation) and inner (training) data. $Y_1 \in \mathbb{R}^{n_1}$ and $Y_2 \in \mathbb{R}^{n_2}$ are the corresponding label vectors, respectively.

Intuitively, the inner problem in eq. (9) finds the set $\mathcal{S}_\Lambda$ of best model parameters $w^*$, and the upper problem optimizes $\Lambda$ so that the *worst* performing $w^*$ in $\mathcal{S}_\Lambda$ yields minimal validation error. Representations learned this way are robust as they allow all minimizers in $\mathcal{S}_\Lambda$ to achieve low validation error. Note that this problem is intrinsically hard because one needs to compute the set $\mathcal{S}_\Lambda$, which can be intractable. Fortunately, our proposed algorithms AdaProx-SG and AdaProx-PD provide a way to solve problem eq. (9) without having to explicitly find the set $\mathcal{S}_\Lambda$.

In our experiments, we consider regression problems where the loss function $\mathcal{L}(\cdot, \cdot)$ corresponds to the squared $\ell_2$-norm. We conduct the experiments on synthetic random data as in [47]. The input matrices $X_1$ and $X_2$ are well conditioned and Gaussian with zero mean and unit variance. We generate the outputs $Y_1$ and $Y_2$ by applying a linear model on a subset of the features (20% of the features) and adding a random Gaussian noise term.

We plot the experiment results in Figures 2 and 3 in Appendix B due to page limits. Figures 2 and 3 show the performance comparisons between AdaProx-SG and AdaProx-PD w.r.t. the running time for solving the HR problem, when the representation dimension is set to $m = 512$ and $m = 1024$, respectively. As depicted, both algorithms solve the problem within a comparable time frame, while AdaProx-SG is slightly faster. We note the following remarks about the plots in Figures 2 and 3, which are intuitively expected. (a) AdaProx-SG by design tries to minimize the maximum constraint violation and hence is more stable at achieving this goal compared to AdaProx-PD (middle plots in Figures 2 and 3), but this can come with a less stable minimization of the outer objective (left plot in Figure 2). (b) Because AdaProx-SG enforces the constraints more effectively, it also achieves a better optimization of the inner problem, which is just one of the constraints in our reformulation. The fact that AdaProx-SG algorithm is more sensitive to the constraint violations is intuitively expected. Indeed, during the algorithm running, whenever some certain constraints are not satisfied, then AdaProx-SG directly penalizes the maximum violation with no delay in line 10 of Algorithm 2. However, the AdaProx-PD algorithm penalizes the violated constraints through increasing the corresponding Lagrangian terms in $\lambda$, i.e. push the updating direction of $z$ closer to the directions alleviating the violation. We provide the iteration time comparison of AdaProx-SG and AdaProx-PD in Table 2, where AdaProx-SG and AdaProx-PD scale similarly with the problem dimension $m$ and AdaProx-SG is slightly faster.

## 6. Conclusion and Future Work

In this paper, we provide the first-known adaptive proximal point algorithm called AdaProx for pessimistic bilevel optimization. Our algorithm features novel designs of adaptive constraint relaxation and accuracy level in order to guarantee an efficient and provable convergence. We further provide the convergence rate analysis of AdaProx which adopts a standard solvers of SG and PD for solving subproblems, and show that both AdaProx-SG and AdaProx-PD converge to an $\epsilon$-KKT point. Our experiments on an illustrative example and the robust hyper-representation learning problem clearly validate our algorithmic design and theoretical analysis. Moreover, our techniques can also be applied to constrained min-max problems as well as OBO and PBO with functional constraints. For example, suppose PBO has functional constraints in the outer level. The problem can still take the same reformulation as in eq. (3), simply with more additional constraints. Our algorithm and the convergence analysis can still be applied. An interesting direction for future research is establishing a PBO benchmark leveraging SOTA optimistic bilevel algorithms, such as FAST-AT [67] and FAST-BAT [68], applied to the real-world CIFAR-10 dataset.

### Acknowledgement

The work was supported in part by the U.S. National Science Foundation under the grants CCF-1909291, CCF-1900145, ECCS-2413528, and DMS-2134145.

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

# Appendix

## A. Example Solvers for Sub-problems in AdaProx

In this section, we introduce two popular gradient-based methods for constrained convex optimization, which can be used for solving the subproblems in eq. (7) in AdaProx.

### A.1. Switching Gradient (SG) Solver

The switching gradient (SG) method, which was recently proposed for solving constrained convex optimization in [29, 53], can serve as a solver for solving the subproblems in eq. (7) in AdaProx. As illustrated in Algorithm 2, SG features two alternating updates: either updating the variable $z$ along the gradient descent direction of the objective function if all constraints are satisfied (in order to minimize the objective), or updating the variable $z$ along the gradient descent direction of the constraint that has the maximum violation (in order to enforce the constraints).

More specifically, suppose that the variable $z$ is updated as $z_t = (x_t, y_t, w_t)$ at iteration $t$. SG first runs the following gradient descent over $y$ w.r.t. $g_\alpha(x, y)$ as follows:

$$\hat{y}_{n+1}^t = \hat{y}_n^t - \tfrac{2}{L_g + 2\alpha} \left( \nabla_y g(x_t, \hat{y}_n^t) + \alpha \hat{y}_n^t \right), \tag{10}$$

such that $g_\alpha(x, \hat{y}_N^t)$ serves as a good approximation for $g_\alpha^*(x_t) := \max_y g_\alpha(x_t, y)$ in the constraint. We further denote $\hat{h}_k(z_t; \hat{y}_N^t)$ as the approximation of the constraint $h_k(z)$ with $z = z_t$ and $g_\alpha^*(x_t)$ being replaced by $g_\alpha(x, \hat{y}_N^t)$. Next, if the constraint is satisfied, i.e., all components of approximated constraint is small enough $(\max_i \{\hat{h}_k(z_t; \hat{y}_N^t)_i\} \leq \frac{\epsilon}{2}$ for some prescribed $\epsilon > 0)$, then $z_t$ is updated along the gradient descent direction of the objective function $f_k(z_t)$. Otherwise, $z_t$ is updated along the $i_t$-th row of $\widehat{\nabla} h_k(z_t; \hat{y}_N^t)$, where $i_t$ corresponds to the maximum constraint violation component, and $\widehat{\nabla} h_k(z_t; \hat{y}_N^t)$ is the approximation of $\nabla h_k(z)$ where $\nabla h_k(z)$ can be derived based on eq. (6) as:

$$\nabla h_k(z) = \begin{pmatrix} (\nabla_x g(\theta) - \nabla_x g_\alpha^*(x))^\top & (\nabla_y g(\theta))^\top & 0 \\ -\nabla_{yx}^2 f(\theta) + w\nabla_{yx}^2 g(\theta) & -\nabla_{yy}^2 f(\theta) + w\nabla_{yy}^2 g(\theta) & \nabla_y g(\theta) \\ \nabla_{yx}^2 f(\theta) - w\nabla_{yx}^2 g(\theta) & \nabla_{yy}^2 f(\theta) - w\nabla_{yy}^2 g(\theta) & -\nabla_y g(\theta) \\ w \left( \nabla_x g(\theta) - \nabla_x g_\alpha^*(x) \right)^\top & w \left( \nabla_y g(\theta) \right)^\top & g(\theta) - g_\alpha^*(x) - \xi \\ -w \left( \nabla_x g(\theta) - \nabla_x g_\alpha^*(x) \right)^\top & -w \left( \nabla_y g(\theta) \right)^\top & -g(\theta) + g_\alpha^*(x) + \xi \end{pmatrix} + \sigma(z - \tilde{z}_k), \tag{11}$$

where $\theta = (x, y)$ for short. $\widehat{\nabla} h_k(z_t; \hat{y}_N^t)$ is obtained from $\nabla h_k(z_t)$ by replacing $g_\alpha^*(x_t)$ and $\nabla g_\alpha^*(x_t)$ with $g_\alpha(x, \hat{y}_N^t)$ and $\nabla_x g_\alpha(x_t, \hat{y}_N^t)$, respectively.

Note that although the gradient of $\nabla h(z)$ in eq. (11) involves the calculation of the second-order Jacobian and Hessian terms of $f$ and $g$, the computational complexity is not demanding since each update uses only one row of the matrix.

---

**Algorithm 2** Switching Gradient (SG) Solver

---

1: **Input:** Number of iterations $T$ and $N$, stepsizes $\{\gamma_t\}_{t=0}^{T-1}$, violation tolerance $\epsilon$
2: Initialize feasible indices set $\mathcal{T} = \emptyset$ and $z_0 \in \mathcal{Z}$
3: **for** $t = 1, ..., T$ **do**
4:   Conduct projected gradient descent in eq. (10) for $N$ times with any given $\hat{y}_0^t$ as initialization
5:   **if** $\max_j \left\{ \left( \hat{h}_k(z_t; \hat{y}_N^t) \right)_j \right\} \leq \frac{\epsilon}{2}$ **then**
6:     $\mathcal{T} = \mathcal{T} \cup \{t\}$
7:     $z_{t+1} = \Pi_{\mathcal{Z}} \left( z_t - \gamma_t^{-1} \nabla f_k(z_t) \right)$
8:   **else**
9:     Let $i_t = \arg\max_j \left\{ \left( \hat{h}_k(z_t; \hat{y}_N^t) \right)_j \right\}$.
10:     $z_{t+1} = \Pi_{\mathcal{Z}} \left( z_t - \gamma_t^{-1} \left( \widehat{\nabla} h_k(z_t; \hat{y}_N^t) \right)_{i_t} \right)$
11:   **end if**
12: **end for**
13: **Output:** $\tilde{z}_{k+1} = \sum_{t \in \mathcal{T}} \gamma_t z_t / \left( \sum_{t \in \mathcal{T}} \gamma_t \right)$

---

## A.2. Primal-Dual (PD) Solver

As a standard method for solving constrained convex optimization, the primal-dual (PD) method can also serve as a solver for solving the subproblems in eq. (7) in AdaProx. Specifically, PD solver in Algorithm 3 solves the minimax problem over the Lagrangian function defined below:

$$\min_{z \in \mathcal{Z}} \max_{\lambda \in \mathbb{R}_+^p} \mathcal{L}_k(z, \lambda) := f_k(z) + \langle h_k(z), \lambda \rangle, \tag{12}$$

where $\lambda \in \mathbb{R}_+^p$ is the dual variable, by alternatively updating the primal variable $z$ and the dual variable $\lambda$ through gradient descent and gradient ascent, respectively. Because the gradients $\nabla_z \mathcal{L}_k(z, \lambda) = \nabla_z f_k(z) + (\nabla_z h_k(z))^\top \lambda$ and $\nabla_\lambda \mathcal{L}_k(z, \lambda) = h_k(z)$, we also need to run a subroutine to estimate $h_k(z)$ and $\nabla_z h_k(z)$, as what we have done in eq. (10). Then, the estimations of $\nabla_z \mathcal{L}_k(z, \lambda)$ and $\nabla_\lambda \mathcal{L}_k(z, \lambda)$ at the iterate $(z_t, \lambda_{t+1})$ immediately follow as: $\widehat{\nabla}_z \mathcal{L}_k(z_t, \lambda_{t+1}; \hat{y}_N^t) = \nabla_z f_k(z_t) + (\widehat{\nabla}_z h_k(z_t; \hat{y}_N^t))^\top \lambda_{t+1}$ and $\widehat{\nabla}_\lambda \mathcal{L}_k(z_t, \lambda_{t+1}) = \hat{h}_k(z_t; \hat{y}_N^t)$.

We then conduct the accelerated gradient ascent and gradient descent to the Lagrangian:

$$\lambda_{t+1} = \Pi_\Lambda \left( \lambda_t + \frac{1}{\eta_t} \left( (1 + \theta_t) \hat{h}_k(z_t; \hat{y}_N^t) - \theta_t \hat{h}_k(z_{t-1}; \hat{y}_N^{t-1}) \right) \right), \tag{13}$$

$$z_{t+1} = \Pi_{\mathcal{Z}} \left( z_t - \frac{1}{\tau_t} \widehat{\nabla}_z \mathcal{L}_k(z_t, \lambda_{t+1}; \hat{y}_N^t) \right), \tag{14}$$

where $\tau_t, \eta_t$ are the stepsizes, $\theta_t$ is the momentum weight, and $\Lambda \subseteq \mathbb{R}_+$ is a closed and bounded set.

---

**Algorithm 3** Primal-Dual (PD) Solver

---

1: **Input:** stepsizes $\eta_t, \tau_t$, momentum weights $\theta_t$, output weight $\gamma_t$, initialization $z_0, \lambda_0$, and iteration times $T$ and $N$
2: **for** $t = 0, 1, ..., T - 1$ **do**
3:   Conduct projected gradient descent in eq. (10) for $N$ times with any given $\hat{y}_0^t$ as initialization
4:   Update $\lambda_{t+1}$ according to eq. (13)
5:   Update $z_{t+1}$ according to eq. (14)
6: **end for**
7: **Output:** $\tilde{z}_{k+1} = \frac{1}{\Gamma_T} \sum_{t=0}^{T-1} \gamma_t z_{t+1}$, where $\Gamma_T = \sum_{t=0}^{T-1} \gamma_t$

---

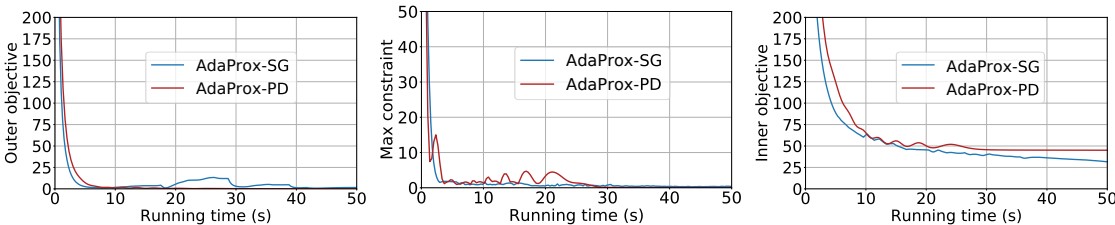

Figure 2: Comparison of AdaProx-PD and AdaProx-SG for the robust HR problem in eq. (9) with $m = 512$

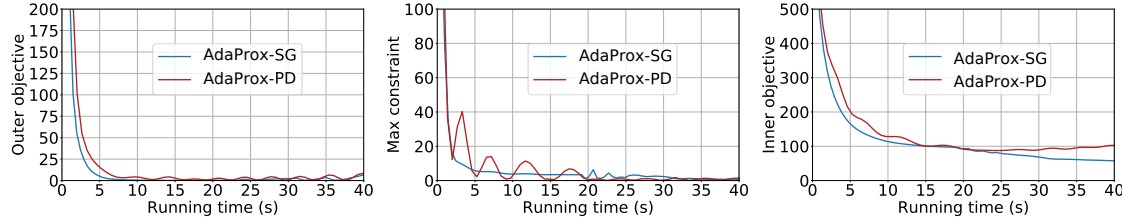

Figure 3: Comparison of AdaProx-PD and AdaProx-SG for the robust HR problem in eq. (9) with $m = 1024$

# B. Figures of Learning Robust Hyper-representation Experiment in Section 5.2

In this section, we provide the figures in Figures 2 and 3 for the learning robust hyper-representation experiment in Section 5.2.

# C. Proof of Proposition 1

## C.1. Supporting Lemmas

**Lemma 1.** *For any given $x \in \mathcal{X}$, consider the following constrained optimization problem.*

$$\min_{y \in \mathbb{R}^m} \quad -f(x, y)$$
$$s.t. \quad g(x, y) - g_\alpha^*(x) - \xi \leq 0. \tag{15}$$

*There exists $y^*(x) \in \mathcal{Y}$ that attains the solution of the above problem. Moreover, there exists $w^*(x) \geq 0$, such that the following KKT condition holds.*

$$-\nabla_y f(x, y^*(x)) + w^*(x)\nabla_y g(x, y) = 0$$
$$w^*(x)g(x, y^*(x)) - g_\alpha^*(x) - \xi) = 0. \tag{16}$$

*For all $w^*(x)$ satisfying the above KKT condition, we have $w^*(x) \leq \frac{\Delta_f}{\xi}$ with*

$$\Delta_f := \max_{x, x' \in \mathcal{X}, y, y' \in \mathcal{Y}} |f(x, y) - f(x', y')|.$$

*Proof.* Given $x \in \mathcal{X}$, let $\tilde{y} \in \mathcal{S}(x)$. We have $g(x, \tilde{y}) - g_\alpha^*(x) - \xi \leq -\xi$. Thus, $\tilde{y}$ is a strictly feasible point with margin $\xi$ for the problem in eq. (15).

Define the dual function $d(w) = \min_{y \in \mathbb{R}^m} -f(x, y) + w(g(x, y) - g_\alpha^*(x) - \xi)$. By its definition, we have, for any $w \in \mathbb{R}_+$ and $y \in \mathbb{R}^m$,

$$d(w) \leq -f(x, \tilde{y}) + w(g(x, y) - g_\alpha^*(x) - \xi) = -f(x, \tilde{y}) - w\xi. \tag{17}$$

Moreover, it is known that convex constrained optimization has no duality gap [69]. And the existence of $\tilde{y}$ ensures the Slater's condition holds. Therefore, the existence of $y^*(x)$ and $w^*(x)$ is

ensured. And, eq. (16) is the necessary and sufficient condition for the optimality of eq. (15). In the other words, $d(w^*(x)) = d^* = p^* = -f(x, y^*(x))$. Taking $w = w^*(x)$ in eq. (17), we obtain

$$-f(x, y^*(x)) = d(w^*(x)) \leq -f(x, \tilde{y}) - w^*(x)\xi.$$

Rearranging terms in the above inequality, we have

$$w^*(x) \leq \frac{f(x, y^*(x)) - f(x, \tilde{y})}{\xi} \overset{(i)}{\leq} \frac{\Delta_f}{\xi}.$$

where $(i)$ follows from the definition of $\Delta_f$. □

Then, we propose the following proposition that provide the clear description of equivalence between eqs. (4) and (5).

**Lemma 2.** *The minimax problem eq.* (4) *is equivalent to the following constrained optimization:*

$$\min_{z \in \mathcal{Z}} \quad f(z)$$

$$s.t. \quad h(z) := \begin{pmatrix} g(x, y) - g_\alpha^*(x) - \xi \\ -\nabla_y f(x, y) + w \nabla_y g(x, y) \\ \nabla_y f(x, y) - w \nabla_y g(x, y) \\ w(g(x, y) - g_\alpha^*(x) - \xi) \\ -w(g(x, y) - g_\alpha^*(x) - \xi) \end{pmatrix} \leq 0, \tag{18}$$

*where* $z = (x, y, z)$, $\mathcal{W} := [0, \frac{\Delta_f}{\xi}]$, *with* $\Delta_f := \max_{x, x' \in \mathcal{X}, y, y' \in \mathcal{Y}} |f(x, y) - f(x', y')|$, *and* $\mathcal{Y} := \{y \in \mathbb{R}^m : \|y\|_2 \leq D_{\mathcal{Y}}\}$ *with* $D_{\mathcal{Y}} > 0$, *such that, for all* $x \in \mathcal{X}$, $\{y \in \mathbb{R}^m : g(x, y) - g_\alpha^*(x) - \xi \leq 0\} \subseteq \mathcal{Y}$, $\mathcal{Z} = \mathcal{X} \times \mathcal{Y} \times \mathcal{W}$, *and* $f(z) = f(x, y)$.

*Proof.* Let $p^* = \min_{x \in \mathcal{X}} \{\psi(x) := \max_{y \in \mathbb{R}^m} \{f(x, y) : g(x, y) - g_\alpha^*(x) - \xi \leq 0\}\}$ be the solution of eq. (4). And let $p_r^* = \min_{x \in \mathcal{X}} \psi_r(x)$ be the solution of eq. (18), with

$$\psi_r(x) := \min_{y \in \mathcal{Y}, w \in \mathcal{W}} \quad f(x, y)$$

$$s.t. \quad g(x, y) - g_\alpha^*(x) - \xi \leq 0$$

$$- \nabla_y f(x, y) + w \nabla_y g(x, y) = 0$$

$$w(g(x, y) - g_\alpha^*(x) - \xi) = 0. \tag{19}$$

By Lemma 1, the feasible set for a given $x \in \mathcal{X}$ of eq. (19) is non-empty, i.e., there exist at least one $(y^*(x), w^*(x)) \in \mathcal{Y} \times \mathcal{W}$ satisfying all three constraints, which implies $\psi_r(x) \leq +\infty$. Moreover, for all $(y, w)$ in the feasible set of eq. (19), we have it satisfies the KKT condition and $g(x, y) - g_\alpha^*(x) - \xi \leq 0$, which the sufficient condition for $y$ to be the solution of eq. (4), i.e., $f(x, y) = \psi(x)$. Therefore, we have $\psi(x) = \psi_r(x)$ for all $x \in \mathcal{X}$, which complete the proof. □

## C.2. Proof of Proposition 1

By Lemma 2 and Assumption 2, we have an equivalent expression of $\Phi_{\alpha, \xi}(x)$ as

$$\Phi_{\alpha, \xi}(x) = \max_{y \in \mathbb{R}^m} \{f(x, y) : g(x, y) - g_\alpha^*(x) - \xi \leq 0\}.$$

Given an $x \in \mathcal{X}$, it is clear that

$$\{y \in \mathbb{R}^m : g(x, y) - g^*(x) \leq 0\} \subseteq \{y \in \mathbb{R}^m : g(x, y) - g_\alpha^*(x) - \xi \leq 0\}.$$

Thus, we have

$$\Phi(x) = \max_{y \in \mathbb{R}^m} \{f(x, y) : g(x, y) - g^*(x) \leq 0\} \leq \max_{y \in \mathbb{R}^m} \{f(x, y) : g(x, y) - g_\alpha^*(x) - \xi \leq 0\} = \Phi_{\alpha, \xi}(x). \tag{20}$$

Moreover, suppose $y^*(x) \in \{y \in \mathbb{R}^m : g(x, y) - g^*(x) \leq 0\}$ and $y_{\alpha, \xi}^*(x) \in \{y \in \mathbb{R}^m : g(x, y) - g_\alpha^*(x) - \xi \leq 0\}$ satisfying $f(x, y^*(x)) = \Phi(x)$ and $f(x, y_{\alpha, \xi}^*(x)) = \Phi_{\alpha, \xi}(x)$. Then, there exist two conditions:

($a$). Suppose $y^*_{\alpha,\xi}(x) \in \{y \in \mathbb{R}^m : g(x,y) - g^*(x) \le 0\}$. Then, by the definition of $\Phi(x)$, we have

$$\Phi_{\alpha,\xi}(x) = f(x, y^*_{\alpha,\xi}(x)) \le \max_{y \in \mathbb{R}^m} \{f(x,y) : g(x,y) - g^*(x) \le 0\} = \Phi(x). \tag{21}$$

($b$). Suppose $y^*_{\alpha,\xi}(x) \notin \{y \in \mathbb{R}^m : g(x,y) - g^*(x) \le 0\}$. Because $g(x,y)$ is convex on $y$, $\mathcal{S}(x) = \{y \in \mathbb{R}^m : g(x,y) - g^*(x) \le 0\}$ is a convex set. Let $\tilde{y}$ be the orthogonal projection of $y^*_{\alpha,\xi}(x)$ on $\mathcal{S}(x)$. Since $\tilde{y} \in \mathcal{S}(x)$, we have

$$
\begin{aligned}
g(x, y^*_{\alpha,\xi}(x)) - g^*(x) &= g(x, y^*_{\alpha,\xi}(x)) - g(x, \tilde{y}) \\
&= \int_{t=0}^{1} \langle \nabla_y g(x, \tilde{y} + t(y^*_{\alpha,\xi}(x) - \tilde{y})), y^*_{\alpha,\xi}(x) - \tilde{y} \rangle dt \\
&= \int_{t=0}^{1} \left\langle \int_{s=0}^{t} \nabla^2_{yy} g(x, \tilde{y} + s(y^*_{\alpha,\xi}(x) - \tilde{y})) ds, y^*_{\alpha,\xi}(x) - \tilde{y} \right\rangle dt \\
&= \int_{t=0}^{1} \int_{s=0}^{t} (y^*_{\alpha,\xi}(x) - \tilde{y})^\top \nabla^2_{yy} g(x, \tilde{y} + s(y^*_{\alpha,\xi}(x) - \tilde{y}))(y^*_{\alpha,\xi}(x) - \tilde{y}) ds\, dt \\
&\overset{(i)}{\ge} \tfrac{\kappa}{2} \|y^*_{\alpha,\xi}(x) - \tilde{y}\|_2^2, \tag{22}
\end{aligned}
$$

where ($i$) follows from the facts that, for any $s \in [0,t] \subseteq [0,1]$, $\nabla_y g(x, y(s)) \ne 0$, where we denote $y(s) := \tilde{y} + s(y^*_{\alpha,\xi}(x) - \tilde{y})$ for short, and thus

$$(y^*_{\alpha,\xi}(x) - \tilde{y})^\top \nabla^2_{yy} g(x, y(s))(y^*_{\alpha,\xi}(x) - \tilde{y}) \ge \lambda_{min}(\nabla^2_{yy} g(x, y(s)) \|y^*_{\alpha,\xi}(x) - \tilde{y}\|_2^2 > \kappa \|y^*_{\alpha,\xi}(x) - \tilde{y}\|_2^2.$$

Moreover, it is clear that

$$g(x, y^*_{\alpha,\xi}(x)) \overset{(i)}{\le} g^*_\alpha(x) + \xi \overset{(ii)}{\le} g^*(x) + \tfrac{\alpha}{2} D_{\mathcal{Y}}^2 + \xi, \tag{23}$$

where ($i$) follows from the fact that $y^*_{\alpha,\xi}(x) \in \{y \in \mathbb{R}^m : g(x,y) - g^*_\alpha(x) - \xi \le 0\}$, and ($ii$) follows from $g^*_\alpha(x) \le g_\alpha(x, y^*(x)) = g(x, y^*(x)) + \tfrac{\alpha}{2}\|y^*(x)\|_2^2 \le g^*(x) + \tfrac{\alpha}{2} D_{\mathcal{Y}}^2$.

Combining eqs. (22) and (23), we obtain

$$\|y^*_{\alpha,\xi}(x) - \tilde{y}\|_2 \le \sqrt{\tfrac{2}{\kappa}\left(\tfrac{D_{\mathcal{Y}}^2}{2}\alpha + \xi\right)}. \tag{24}$$

By the Lipschitz continuity of $f(x,y)$, there exists $M > 0$ such that

$$
\begin{aligned}
f(x, y^*_{\alpha,\xi}(x)) &\le f(x, \tilde{y}) + M\|y^*_{\alpha,\xi}(x) - \tilde{y}\|_2 \\
&\overset{(i)}{=} f(x, y^*(x)) + M\|y^*_{\alpha,\xi}(x) - \tilde{y}\|_2 \\
&\overset{(ii)}{\le} f(x, y^*(x)) + M\sqrt{\tfrac{2}{\kappa}\left(\tfrac{D_{\mathcal{Y}}^2}{2}\alpha + \xi\right)}, \tag{25}
\end{aligned}
$$

where ($i$) follows from $\tilde{y} \in \{y \in \mathbb{R}^n : g(x,y) - g^*(x) \le 0\}$, and ($ii$) follows from eq. (24).

Equation (25) implies that $\Phi_{\alpha,\xi}(x) \le \Phi(x) + \mathcal{O}(\sqrt{\xi}) + \mathcal{O}(\sqrt{\alpha})$. Together with eqs. (20) and (21), we complete the proof.

## D. Proof Proposition 2

We first provide the Lipschitz condition lemma as follows.

**Lemma 3.** *Given a function $J : \mathbb{R}^n \to \mathbb{R}$, which is twice differentiable and is a $L_J$-gradient Lipschitz function on the bounded support $\mathcal{X} \subseteq \mathbb{R}^n$, and for all $x \in \mathcal{X}$, $\|\nabla J(x)\|_2 \le M_J$. Then, define a new function $I : \mathcal{X} \times [0, B] \to \mathbb{R}$ as $I(x,y) = yJ(x)$. We have $I(x,y)$ is a $(BL_J + M_J)$- gradient Lipschitz function.*

*Proof.* By the definition of $I(x,y)$, we have its gradient $\nabla I(x,y) = [\nabla_x I(x,y); \frac{\partial I(x,y)}{\partial y}]$ equals $[y\nabla_x J(x); J(x)]$. And its Hessian equals

$$\nabla^2 I(x,y) = \begin{pmatrix} y\nabla^2_{xx}J(x) & \nabla_x J(x) \\ (\nabla_x J(x))^\top & 0 \end{pmatrix},$$

where we let $\nabla^2 = \nabla^2_{(x,y),(x,y)}$.

Let $z = (a,b)^\top \in \mathbb{R}^n$, with $a \in \mathbb{R}^n$ and $b \in \mathbb{R}$, for any $x \in \mathcal{X}$ and $0 \le y \le B$, we have

$$\begin{aligned}
z^\top \nabla^2 I(x,y)z &= ya^\top \nabla^2_{xx}J(x)a + 2b \cdot a^\top \nabla_x J(x) \\
&\overset{(i)}{\le} yL_J\|a\|_2^2 + 2b\|a\|_2\|\nabla_x J(x)\|_2 \\
&\overset{(ii)}{\le} yL\|z\|_2^2 + (\|a\|_2^2 + b^2)\|\nabla_x J(x)\|_2 \\
&\le (BL_J + M_J)\|z\|_2^2,
\end{aligned}$$

where $(i)$ follows from the $L_J$ gradient Lipschitz condition of $J(x)$ and Cauchy-Schwartz inequality and $(ii)$ follows from the Young's inequality. $\qquad\square$

In the following proof, we consider each component of the $h(z)$ and prove that they are $L_c$ gradient Lipschitz, with

$$L_c = \max\left\{ 2\rho_f + \frac{4\mu_g \Delta_f}{\xi} + 4M_{Hg} + L_g, \frac{\Delta_f}{\xi}(2L_g + \frac{L_g^2}{\alpha}) + M_g \right\},$$

where $M_g = \sup_{z \in \mathcal{Z}} \|\nabla_z(g(x,y) - g_\alpha^*(x))\|_2$ and $M_{Hg} = \sup_{z \in \mathcal{Z}} \|\nabla^2_{zz}g(z)\|_F$.

For the first component $g(x,y) - g_\alpha^*(x) - \xi$, it has been shown to be $(2L_g + L_g^2/\alpha)$-gradient Lipschitz (Lemma 1 of [15]). The next $m$ components of $h(z)$ are the entries of $-\nabla_y f(x,y) + w\nabla_y g(x,y)$. Consider the $i$th entry. For any given $z$ and $z' \in \mathcal{Z}$, let $e_i^+(z) = (-\nabla_y f(x,y) + w\nabla_y g(x,y))_i$, we have

$$\begin{aligned}
&\|\nabla e_i^+(z) - \nabla e_i^+(z')\|_2^2 \\
&= \left\| \nabla\left(-\nabla_y f(x,y) + w\nabla_y g(x,y)\right)_i - \nabla\left(-\nabla_y f(x',y') + w'\nabla_y g(x',y')\right)_i \right\|_2^2 \\
&\overset{(i)}{=} \left\| \left(-\nabla^2_{yx}f(x,y) + w\nabla^2_{yx}g(x,y)\right)_{(i,\cdot)} - \left(-\nabla^2_{yx}f(x',y') + w'\nabla^2_{yx}g(x',y')\right)_{(i,\cdot)} \right\|_2^2 \\
&\quad + \left\| \left(-\nabla^2_{yy}f(x,y) + w\nabla^2_{yy}g(x,y)\right)_{(i,\cdot)} - \left(-\nabla^2_{yy}f(x',y') + w'\nabla^2_{yy}g(x',y')\right)_{(i,\cdot)} \right\|_2^2 \\
&\quad + \left((\nabla_y g(x,y))_i - (\nabla_y g(x',y'))_i\right)^2 \\
&\overset{(ii)}{\le} 2\left\| \left(\nabla^2_{yx}f(x,y) - \nabla^2_{yx}f(x',y')\right)_{(i,\cdot)} \right\|_2^2 \\
&\quad + 2\left\| \left(\nabla^2_{yx}g(x,y) - \nabla^2_{yx}g(x',y')\right)_{(i,\cdot)} + (w-w')\left(\nabla^2_{yx}g(x',y')\right)_{(i,\cdot)} \right\|_2^2 \\
&\quad + 2\left\| \left(\nabla^2_{yy}f(x,y) - \nabla^2_{yy}f(x',y')\right)_{(i,\cdot)} \right\|_2^2 \\
&\quad + 2\left\| \left(\nabla^2_{yy}g(x,y) - \nabla^2_{yy}g(x',y')\right)_{(i,\cdot)} + (w-w')\left(\nabla^2_{yy}g(x',y')\right)_{(i,\cdot)} \right\|_2^2 \\
&\quad + \left((\nabla_y g(x,y))_i - (\nabla_y g(x',y'))_i\right)^2 \qquad\qquad (26)
\end{aligned}$$

where $(i)$ follows from $\|\nabla_z h\|_2^2 = \|\nabla_x h\|_2^2 + \|\nabla_y h\|_2^2 + (\frac{\partial h}{\partial w})^2$ and $(M)_{(i,\cdot)}$ denotes the $i$th row of the matrix $M$, and $(ii)$ follows from the fact $\|a+b\|_2^2 \le 2\|a\|_2^2 + 2\|b\|_2^2$.

Using the fact that $\sqrt{a+b} \le \sqrt{a} + \sqrt{b}$ for all $a,b \ge 0$, eq. (26) induces

$$\begin{aligned}
&\|\nabla e_i^+(z) - \nabla e_i^+(z')\|_2 \\
&\le 2\left\| \left(\nabla^2_{yx}f(x,y) - \nabla^2_{yx}f(x',y')\right)_{(i,\cdot)} \right\|_2 \\
&\quad + 4\left\| v\left(\nabla^2_{yx}g(x,y) - \nabla^2_{yx}g(x',y')\right)_{(i,\cdot)} \right\|_2 + 4\left\| (w-w')\left(\nabla^2_{yx}g(x',y')\right)_{(i,\cdot)} \right\|_2
\end{aligned}$$

$$+ 2 \left\| \left( \nabla^2_{yy} f(x,y) - \nabla^2_{yy} f(x',y') \right)_{(i,\cdot)} \right\|_2$$
$$+ 4 \left\| w \left( \nabla^2_{yy} g(x,y) - \nabla^2_{yy} g(x',y') \right)_{(i,\cdot)} \right\|_2 + 4 \left\| (w - w') \left( \nabla^2_{yy} g(x',y') \right)_{(i,\cdot)} \right\|_2$$
$$+ \left| (\nabla_y g(x,y))_i - (\nabla_y g(x',y'))_i \right|$$
$$\le \left( 2\rho_f + \tfrac{4\mu_g \Delta_f}{\xi} + 4M_{Hg} + L_g \right) \|z - z'\|_2, \tag{27}$$

where $M_{Hg} = \sup_{z \in \mathcal{Z}} \|\nabla^2 g\|_F$.

Next, let $e_i^-(z) = (\nabla_y f(x,y) - w \nabla_y g(x,y))_i$. Following the same steps in eqs. (26) and (27), we also obtain

$$\|\nabla e_i^-(z) - \nabla e_i^-(z')\|_2 \le \left( 2\mu_f + \tfrac{4\mu_g \Delta_f}{\xi} + 4M_{Hg} + L_g \right) \|z - z'\|_2 \tag{28}$$

For the last two components, $w(g(x,y) - g_\alpha^*(x) - \xi)$ and $-w(g(x,y) - g_\alpha^*(x) - \xi)$, because $g(x,y) - g_\alpha^*(x)$ is $(2L_g + \tfrac{L_g^2}{\alpha})$-gradient Lipschitz. Moreover, since the support $\mathcal{Z}$ is bounded, there exist $M_g$, such that $\|\nabla(g(x,y) - g_\alpha^*(x) - \xi)\|_2 \le M_g$, and $w$ is bounded in interval $[0, \tfrac{\Delta_f}{\xi}]$. Applying Lemma 3, we have $w(g(x,y) - g_\alpha^*(x) - \xi)$ and $-w(g(x,y) - g_\alpha^*(x) - \xi)$ are $\tfrac{\Delta_f}{\xi}(2L_g + \tfrac{L_g^2}{\alpha}) + M_g$ gradient Lipschitz.

## E. Proof of Theorem 1

**Lemma 4** (Theorem 2.2.14 [54]). *Suppose Assumption 1 holds. Consider the gradient descent in eq. (10). We have*

$$\|\hat{y}_N^t - y_\alpha^*(x_t)\|_2 \le \left( 1 - \tfrac{\alpha}{L_g + 2\alpha} \right)^N \|\hat{y}_0 - \tilde{y}^*(x_t)\|_2.$$

**Lemma 5.** (*Three-point lemma, (Lemma 3.5 of [69])*). *Given $\mathcal{Z} \subseteq \mathbb{R}^q$ is a convex an closed set, let $z_{t+1} = \Pi_{\mathcal{Z}} (z_t - G)$, where $G \in \mathbb{R}^q$. Then, for any point $z \in \mathcal{Z}$, we have*

$$\langle G, z - z_{t+1} \rangle \ge \tfrac{1}{2}\|z - z_{t+1}\|_2^2 + \tfrac{1}{2}\|z_{t+1} - z_t\|_2^2 - \tfrac{1}{2}\|z - z_t\|_2^2.$$

**Lemma 6.** *Suppose Assumption 1 holds. And $\sigma \ge 2\{L_f, L_c\}$. Let $H_k(z) := \max_j \left\{ (h_k(z))_j \right\}$. Consider $i_t$, $\widehat{h}_k(z_t; \hat{y}_N^t)$ and $\widehat{\nabla} h_k(z_t; \hat{y}_N^t)$ specified in Algorithm 2. We have*

$$\left| (\widehat{h}_k(z_t; \hat{y}_N^t))_{i_t} - H_k(z_t) \right| \le (L_g + \alpha) D_{\mathcal{Y}}^2 D_{\mathcal{Z}} \left( 1 - \tfrac{\alpha}{L_g + 2\alpha} \right)^N.$$

*Moreover, let $\widehat{\partial} H_k(z_t) = (\widehat{\nabla} h_k(z_t; \hat{y}_N^t))_{i_t}$, we have for all $z \in \mathcal{Z}$,*

$$H_k(z) \ge H_k(z_t) + \langle \widehat{\partial} H_k(z_t), z - z_t \rangle + \tfrac{\sigma}{4}\|z - z_t\|_2^2 - 4(L_g + \alpha) D_{\mathcal{Y}} D_{\mathcal{Z}}^2 \left( 1 - \tfrac{\alpha}{L_g + 2\alpha} \right)^N.$$

*Proof.* By Proposition 2, we have each entry of $h_k(z)$ is a $\tfrac{\sigma}{2}$-strongly convex function. Moreover, for any given $z \in \mathcal{Z}$, let $I(z) := \arg\max_j \left\{ (h_k(z))_j \right\}$, we have $\nabla(h_k(z))_{I(z)} \in \partial H_k(z)$.

(a). Suppose $I(z_t) = i_t$.

Observing the form of $\hat{h}_k(z_t; \hat{y}_N^t)$, only its first and last two entries do not equal to $h_k(z_t)$. Thus, we have

$$\left| \left( \widehat{h}_k(z_t; \hat{y}_N^t) \right)_{i_t} - H_k(z_t) \right| \le \max \left\{ |g_\alpha(x_t, \hat{y}_N^t) - g_\alpha^*(x_t)|, |w_t(g_\alpha(x_t, \hat{y}_N^t) - g_\alpha^*(x_t))| \right\}$$
$$\overset{(i)}{\le} D_{\mathcal{Z}} |g_\alpha(x_t, \hat{y}_N^t) - g_\alpha^*(x_t)|$$
$$\overset{(ii)}{\le} (L_g + \alpha) D_{\mathcal{Y}} D_{\mathcal{Z}} \|\hat{y}_N^t - y_\alpha^*(x_t)\|_2$$
$$\overset{(iii)}{\le} (L_g + \alpha) D_{\mathcal{Y}}^2 D_{\mathcal{Z}} \left( 1 - \tfrac{\alpha}{L_g + 2\alpha} \right)^N, \tag{29}$$

where $(i)$ follows from $w_t \le D_{\mathcal{Z}}$, $(ii)$ follows from that $g_\alpha(z)$ is $(L_g + \alpha)D_{\mathcal{Y}}$ Lipschitz continuous, and $(iii)$ follows from Lemma 4.

It is clear that $\widehat{\partial} H_k(z_t) - \partial H_k(z_t) \ne 0$ if and only if $i_t$ selects the first or the last two constraints, i.e., $\|\widehat{\partial} H_k(z_t) - \partial H_k(z_t)\|_2$ equals one of the following three: $0$, $\|((\nabla_x g_\alpha(x_t, \hat{y}_N^t) - \nabla_x g_\alpha^*(x))^\top, 0, 0)\|_2$, or $\|(w_t(\nabla_x g_\alpha(x_t, \hat{y}_N^t) - \nabla_x g_\alpha^*(x_t))^\top, 0, g_\alpha(x_t, \hat{y}_N^t) - g_\alpha^*(x_t))\|_2$. Thus, we have

$$\|\widehat{\partial} H_k(z_t) - \partial H_k(z_t)\|_2 \le \sqrt{w_t^2 \|\nabla_x g_\alpha(x_t, \hat{y}_N^t) - \nabla_x g_\alpha^*(x_t)\|_2^2 + \|g_\alpha(x_t, \hat{y}_N^t) - g_\alpha^*(x_t)\|_2^2}$$

$$\overset{(i)}{\le} w_t \|\nabla_x g_\alpha(x_t, \hat{y}_N^t) - \nabla_x g_\alpha^*(x_t)\|_2 + \|g_\alpha(x_t, \hat{y}_N^t) - g_\alpha^*(x_t)\|_2$$

$$\overset{(ii)}{\le} D_{\mathcal{Z}}(L_g + \alpha)\|y_\alpha^*(x_t) - \hat{y}_N^t\|_2 + (L_g + \alpha)D_{\mathcal{Z}}\|y_\alpha^*(x_t) - \hat{y}_N^t\|_2$$

$$\overset{(iii)}{\le} 2(L_g + \alpha)D_{\mathcal{Z}}D_{\mathcal{Y}}\left(1 - \tfrac{\alpha}{L_g + 2\alpha}\right)^N, \tag{30}$$

where $(i)$ follows from the $\sqrt{x+y} \le \sqrt{x} + \sqrt{y}$ for $x, y \ge 0$, $(ii)$ follows from $\nabla_x g(x, y)$ is $L_g + \alpha$ gradient Lipschitz, $w_t \le D_{\mathcal{Z}}$, and $g_\alpha(x, y)$ is $(L_g + \alpha)D_{\mathcal{Z}}$ Lipschitz continuous, and $(iii)$ follows from Lemma 4. Following the definition of $\partial H_k(z_t)$, strong convexity, and Cauchy Schwartz inequality, we obtain

$$H_k(z) \ge H_k(z_t) + \langle \widehat{\partial} H_k(z_t), z - z_t \rangle + \tfrac{\sigma}{4}\|z - z_t\|_2^2 - 2(L_g + \alpha)D_{\mathcal{Z}}^2 D_{\mathcal{Y}}\left(1 - \tfrac{\alpha}{L_g + 2\alpha}\right)^N. \tag{31}$$

(b). Suppose $I(z_t) \ne i_t$.

Similar to eq. (29), we have $\left|(\widehat{h}_k(z_t; \hat{y}_N^t))_{i_t} - (h_k(z_t))_{i_t}\right| \le (L_g + \alpha)D_{\mathcal{Y}}^2 D_{\mathcal{Z}}\left(1 - \tfrac{\alpha}{L_g + 2\alpha}\right)^N$ and

$$\left|(\widehat{h}_k(z_t; \hat{y}_N^t))_{I(z_t)} - H_k(z_t)\right| \le (L_g + \alpha)D_{\mathcal{Y}}^2 D_{\mathcal{Z}}\left(1 - \tfrac{\alpha}{L_g + 2\alpha}\right)^N.$$

Together with the facts that $(\widehat{h}_k(z_t; \hat{y}_N^t))_{I(z)} \le (\widehat{h}_k(z_t; \hat{y}_N^t))_{i_t}$ and $H_k(z_t) \ge (h_k(z_t))_{i_t}$, we have

$$\left|(\widehat{h}_k(z_t; \hat{y}_N^t))_{i_t} - H_k(z_t)\right| \le (L_g + \alpha)D_{\mathcal{Y}}^2 D_{\mathcal{Z}}\left(1 - \tfrac{\alpha}{L_g + 2\alpha}\right)^N \tag{32}$$

$$H_k(z_t) - 2(L_g + \alpha)D_{\mathcal{Y}}^2 D_{\mathcal{Z}}\left(1 - \tfrac{\alpha}{L_g + 2\alpha}\right)^N \le (h(z_t))_{i_t} \le H_k(z_t). \tag{33}$$

Given $z \in \mathcal{Z}$, following the strong convexity of $(h_k(z))_{i_t}$, we have

$$H_k(z) \ge (h_k(z))_i \ge (h_k(z_t))_{i_t} + \langle \nabla(h_k(z_t))_{i_t}, z - z_t \rangle + \tfrac{\sigma}{4}\|z - z_t\|_2^2$$

$$\overset{(i)}{\ge} H_k(z_t) - 2(L_g + \alpha)D_{\mathcal{Y}}^2 D_{\mathcal{Z}}\left(1 - \tfrac{\alpha}{L_g + 2\alpha}\right)^N + \langle \widehat{\partial} H_k(z_t), z - z_t \rangle + \tfrac{\sigma}{2}\|z - z_t\|_2^2$$

$$+ \langle \nabla(h_k(z_t))_{i_t} - \widehat{\partial} H_k(z_t), z - z_t \rangle$$

$$\overset{(ii)}{\ge} H_k(z_t) + \langle \widehat{\partial} H_k(z_t), z - z_t \rangle + \tfrac{\sigma}{4}\|z - z_t\|_2^2 - 4(L_g + \alpha)D_{\mathcal{Y}}D_{\mathcal{Z}}^2\left(1 - \tfrac{\alpha}{L_g + 2\alpha}\right)^N, \tag{34}$$

where $(i)$ follows from eq. (33) and $(ii)$ follow from eq. (30), Cauchy-Schwartz inequality and $D_{\mathcal{Y}} \le D_{\mathcal{Z}}$.

Thus, from eqs. (29) and (32), we conclude

$$\left|(\widehat{h}_k(z_t; \hat{y}_N^t))_{i_t} - H_k(z_t)\right| \le (L_g + \alpha)D_{\mathcal{Y}}^2 D_{\mathcal{Z}}\left(1 - \tfrac{\alpha}{L_g + 2\alpha}\right)^N.$$

From eqs. (31) and (34), we conclude

$$H_k(z) \ge H_k(z_t) + \langle \widehat{\partial} H_k(z_t), z - z_t \rangle + \tfrac{\sigma}{4}\|z - z_t\|_2^2 - 4(L_g + \alpha)D_{\mathcal{Y}}D_{\mathcal{Z}}^2\left(1 - \tfrac{\alpha}{L_g + 2\alpha}\right)^N.$$

$\square$

**Theorem 4** (Formal Statement of Theorem 1). *Suppose Assumption 1 holds. Consider Algorithm 2. Let $\sigma = \max\{2L_f, 2L_c\}$, $\gamma_t = \frac{\sigma(t+1)}{2}$, $T \geq \frac{4M^2}{\sigma\epsilon}$, with $M = \sup_{z \in D_{\mathcal{Z}}} \|\nabla f_k(z)\|$, and $N \geq \log\left(\frac{\epsilon}{4(T+2)^2(L_g+\alpha)D_{\mathcal{Y}}D_{\mathcal{Z}}^2}\right) / \log(1 - \frac{\alpha}{L_g+2\alpha})$. Then, we have*

$$f_k(\tilde{z}_{k+1}) - f_k(z_k^*) \leq \epsilon, \quad \text{and} \quad \max_j \left\{(h_k(\tilde{z}_{k+1}))_j\right\} \leq \epsilon.$$

*In the other words, we have $\tilde{z}_{k+1}$ is an $\epsilon$-accurate solution of eq. (6).*

*Proof.* Clearly, by the setting of $\sigma$ and proposition 2, we have both $f_k(z)$ and $h_k(z)$ are $\mu = \frac{\sigma}{2}$ strongly convex function. We let $H_k(z) = \max_j \{(h_k(z))_j\}$ for short.

$(a)$. Suppose $t \in \mathcal{T}$, we have $\hat{h}_k(z_t; \hat{y}_N^t) \leq \frac{\epsilon}{2}$. Applying Lemma 5 to the update with respect to the $\nabla f_k(z)$ ensures that, for any given $z \in \mathcal{Z}$,

$$\gamma_t^{-1}\langle\nabla f_k(z_t), z - z_{t+1}\rangle \geq \tfrac{1}{2}\|z - z_{t+1}\|_2^2 + \tfrac{1}{2}\|z_{t+1} - z_t\|_2 - \tfrac{1}{2}\|z - z_t\|_2^2. \tag{35}$$

Moreover, using the strongly convexity of $f_k(z)$, we obtain

$$f_k(z_k^*) \geq f_k(z_t) + \langle\nabla f_k(z_t), z_k^* - z_t\rangle + \tfrac{\mu}{2}\|z_k^* - z_t\|_2^2. \tag{36}$$

Taking $z = z_k^*$ in eq. (35) and using eq. (36), we have

$$f_k(z_t) - f_k(z_k^*) \leq \langle\nabla f_k(z_t), z_t - z_{t+1}\rangle - \tfrac{\gamma_t}{2}\|z_{t+1} - z_t\|_2^2 + \tfrac{\gamma_t-\mu}{2}\|z_k^* - z_t\|_2^2 - \tfrac{\gamma_t}{2}\|z_k^* - z_{t+1}\|_2^2$$

$$\overset{(i)}{\leq} \tfrac{\|\nabla f_k(z_t)\|_2^2}{2\gamma_t} + \tfrac{\gamma_t-\mu}{2}\|z_k^* - z_t\|_2^2 - \tfrac{\gamma_t}{2}\|z_k^* - z_{t+1}\|_2^2, \tag{37}$$

where $(i)$ follows from the Young's inequality, $\langle\nabla f_k(z_t), z_t - z_{t+1}\rangle \leq \tfrac{\|\nabla f_k(z_t)\|_2^2}{2\gamma_t} + \tfrac{\gamma_t}{2}\|z_t - z_{t+1}\|_2^2$.

$(b)$. Suppose $t \notin \mathcal{T}$, we have $\hat{h}_k(z_t; \hat{y}_N^t) > \frac{\epsilon}{2}$, Applying Lemma 5 the update with respect to the $\widehat{\nabla}(h_k(z_t; \hat{y}_N^t))_{i_t}$ (we denote as $\widehat{\partial}H_k(z_t)$ for short) ensures that, for any given $z \in \mathcal{Z}$,

$$\gamma_t^{-1}\langle\widehat{\partial}H_k(z), z - z_{t+1}\rangle \geq \tfrac{1}{2}\|z - z_{t+1}\|_2^2 + \tfrac{1}{2}\|z_{t+1} - z_t\|_2 - \tfrac{1}{2}\|z - z_t\|_2^2. \tag{38}$$

Moreover, applying Lemma 6 with $z = z_k^*$, we obtain

$$H_k(z_k^*) \geq H_k(z_t) + \langle\widehat{\partial}H_k(z_t), z_k^* - z_t\rangle + \tfrac{\mu}{2}\|z_k^* - z_t\|_2^2 - 4(L_g+\alpha)D_{\mathcal{Y}}D_{\mathcal{Z}}^2\left(1 - \tfrac{\alpha}{L_g+2\alpha}\right)^N. \tag{39}$$

Take $z = z_k^*$ in eq. (38) and recall eq. (39). We have

$$H_k(z_t) - H_k(z_k^*)$$
$$\leq \langle\widehat{\partial}H_k(z_t), z_t - z_{t+1}\rangle + \tfrac{\gamma_t-\mu}{2}\|z_k^* - z_t\|_2^2 - \tfrac{\gamma_t}{2}\|z_k^* - z_{t+1}\|_2^2 - \tfrac{\gamma_t}{2}\|z_{t+1} - z_t\|_2^2$$
$$\quad + 4(L_g+\alpha)D_{\mathcal{Y}}D_{\mathcal{Z}}^2\left(1 - \tfrac{\alpha}{L_g+2\alpha}\right)^N$$
$$\overset{(i)}{\leq} \tfrac{\|\widehat{\partial}H_k(z_t)\|_2^2}{2\gamma_t} + \tfrac{\gamma_t-\mu}{2}\|z_k^* - z_t\|_2^2 - \tfrac{\gamma_t}{2}\|z_k^* - z_{t+1}\|_2^2 + 4(L_g+\alpha)D_{\mathcal{Y}}D_{\mathcal{Z}}^2\left(1 - \tfrac{\alpha}{L_g+2\alpha}\right)^N, \tag{40}$$

where $(i)$ follows from applying Young's inequality.

Proceeding with the following inductions.

$$\sum_{t\in\mathcal{T}}\gamma_t(f_k(z_t) - f_k(z_k^*)) + \sum_{t\in[T],t\notin\mathcal{T}}\gamma_t H_k(z_t)$$

$$\overset{(i)}{\leq} \sum_{t\in\mathcal{T}}\gamma_t(f_k(z_t) - f_k(z_k^*)) + \sum_{t\in[T],t\notin\mathcal{T}}\gamma_t(H_k(z_t) - H_k(z_k^*))$$

$$\overset{(ii)}{\leq} \sum_{t\in\mathcal{T}}\tfrac{1}{2}\|\nabla f_k(z_t)\|_2^2 + \sum_{t\in[T],t\neq\mathcal{T}}\left(\tfrac{1}{2}\|\widehat{\partial}H_k(z_t)\|_2^2 + 4\gamma_t(L_g+\alpha)D_{\mathcal{Y}}D_{\mathcal{Z}}^2\left(1 - \tfrac{\alpha}{L_g+2\alpha}\right)^N\right)$$

$$+ \sum_{t=1}^{T} \left( \tfrac{\mu^2 (t-1)t}{8} \|z_k^* - z_t\|_2^2 - \tfrac{\mu^2 t(t+1)}{8} \|z_k^* - z_{t+1}\|_2^2 \right)$$

$$= \sum_{t \in \mathcal{T}} \tfrac{1}{2} \|\nabla f_k(z_t)\|_2^2 + \sum_{t \in [T], t \neq \mathcal{T}} \left( \tfrac{1}{2} \|\widehat{\partial} H_k(z_t)\|_2^2 + 2\mu(t+1)(L_g + \alpha) D_{\mathcal{Y}} D_{\mathcal{Z}}^2 \left( 1 - \tfrac{\alpha}{L_g + 2\alpha} \right)^N \right)$$

$$\leq \frac{M^2 T}{2} + \mu(T+2)^2 (L_g + \alpha) D_{\mathcal{Y}} D_{\mathcal{Z}}^2 \left( 1 - \tfrac{\alpha}{L_g + 2\alpha} \right)^N, \tag{41}$$

where $(i)$ follows from $H_k(z_k^*) \leq 0$, and $(ii)$ follows from eqs. (37) and (40).

Recall that, for all $t \in \mathcal{T}$, we have $\hat{h}_k(z_t; \hat{y}_N^t) \leq \tfrac{\epsilon}{2}$. Applying Lemma 6, we have, for all $t \in \mathcal{T}$,

$$H_k(z_t) \leq \frac{\epsilon}{2} + (L_g + \alpha) D_{\mathcal{Y}} D_{\mathcal{Z}}^2 \left( 1 - \tfrac{\alpha}{L_g + 2\alpha} \right)^N. \tag{42}$$

Applying Lemma 6, we have, for all $t \notin \mathcal{T}$,

$$H_k(z_t) \geq \frac{\epsilon}{2} - (L_g + \alpha) D_{\mathcal{Y}} D_{\mathcal{Z}}^2 \left( 1 - \tfrac{\alpha}{L_g + 2\alpha} \right)^N.$$

Multiplying $\gamma_t$ on both sides of the above inequality and telescoping, we obtain

$$\sum_{t \in [T], t \neq \mathcal{T}} \gamma_t H_k(z_t) \geq \sum_{t \in [T], t \notin \mathcal{T}} \gamma_t \left( \epsilon - (L_g + \alpha) D_{\mathcal{Y}}^2 D_{\mathcal{Z}} \left( 1 - \tfrac{\alpha}{L_g + 2\alpha} \right)^N \right)$$

$$\geq \frac{\epsilon}{2} \sum_{t \in [T], t \notin \mathcal{T}} \gamma_t - \mu(T+2)^2 (L_g + \alpha) D_{\mathcal{Y}} D_{\mathcal{Z}}^2 \left( 1 - \tfrac{\alpha}{L_g + 2\alpha} \right)^N.$$

Substituting the above inequality into eq. (41), we obtain

$$\sum_{t \in \mathcal{T}} \gamma_t (f_k(z_t) - f_k(z_k^*)) \leq -\frac{\epsilon}{2} \sum_{t \in [T], t \notin \mathcal{T}} \gamma_t + \frac{M^2 T}{2} + 2\mu(T+2)^2 (L_g + \alpha) D_{\mathcal{Y}} D_{\mathcal{Z}}^2 \left( 1 - \tfrac{\alpha}{L_g + 2\alpha} \right)^N$$

$$\overset{(i)}{\leq} \frac{\epsilon}{2} \sum_{t \in \mathcal{T}} \gamma_t - \frac{\epsilon \mu T^2}{8} + \frac{M^2 T}{2} + 2\mu(T+2)^2 (L_g + \alpha) D_{\mathcal{Y}} D_{\mathcal{Z}}^2 \left( 1 - \tfrac{\alpha}{L_g + 2\alpha} \right)^N,$$

where $(i)$ follows from $-\sum_{t \in [T], t \notin \mathcal{T}} \gamma_t = \sum_{t \in \mathcal{T}} \gamma_t - \sum_{t \in [T]} \gamma_t$ and $\sum_{t \in [T]} \gamma_t \geq \tfrac{\mu T^2}{4}$.

Dividing $\sum_{t \in \mathcal{T}} \gamma_t$ on both side of the above inequality and using the fact $\sum_{t \in \mathcal{T}} \gamma_t \geq \mu$, we obtain

$$\frac{\sum_{t \in \mathcal{T}} \gamma_t (f_k(z_t) - f_k(z_k^*))}{\sum_{t \in \mathcal{T}} \gamma_t} \leq \frac{\epsilon}{2} + \frac{\frac{M^2 T}{2} - \frac{\mu \epsilon T^2}{8}}{\sum_{t \in \mathcal{T}} \gamma_t} + 2(T+2)^2 (L_g + \alpha) D_{\mathcal{Y}} D_{\mathcal{Z}}^2 \left( 1 - \tfrac{\alpha}{L_g + 2\alpha} \right)^N. \tag{43}$$

By the convexity of $f_k(z)$ and eq. (43), we have

$$f_k(\tilde{z}_{k+1}) - f_k(z_k^*) \leq \frac{\epsilon}{2} + \frac{\frac{M^2 T}{2} - \frac{\mu \epsilon T^2}{8}}{\sum_{t \in \mathcal{T}} \gamma_t} + 2(T+2)^2 (L_g + \alpha) D_{\mathcal{Y}} D_{\mathcal{Z}}^2 \left( 1 - \tfrac{\alpha}{L_g + 2\alpha} \right)^N.$$

Finally using the convexity of $H_k(z)$, and eq. (42), we obtain

$$\max_j \left\{ (h_k(\tilde{z}_{k+1}))_j \right\} = H_k(\tilde{z}_{k+1}) \leq \frac{\epsilon}{2} + (L_g + \alpha) D_{\mathcal{Y}} D_{\mathcal{Z}}^2 \left( 1 - \tfrac{\alpha}{L_g + 2\alpha} \right)^N.$$

Recall $N \geq \log \left( \frac{\epsilon}{4(T+2)^2 (L_g + \alpha) D_{\mathcal{Y}} D_{\mathcal{Z}}^2} \right) / \log(1 - \tfrac{\alpha}{L_g + 2\alpha})$ and $T \geq \tfrac{4M^2}{\mu \epsilon}$, we have $f_k(\tilde{z}_{k+1}) - f_k(z_k^*) \leq \epsilon$, and $\max_j \left\{ (h_k(\tilde{z}_{k+1}))_j \right\} \leq \epsilon$. $\qquad \square$

# F. Proof of Theorem 2

Before the proof of Theorem 2, we first prove that the optimal dual variable is upper-bounded.

**Lemma 7.** *Consider the subproblem in eq. (7). When $\sigma \geq 2\{L_f, L_c\}$, we have the optimal dual $\lambda_k^*$ exists and $\|\lambda_k^*\|_1$ satisfies $\|\lambda_k^*\|_1 \leq \frac{f_k(\tilde{z}) - f_k(z_k^*)}{-\max_i\{(h_k(\tilde{z}))_i\}} := B_0$.*

*Proof.* Recall that convex constrained optimization has no duality gap [69]. Then the existence of $\tilde{z}$ ensures that the Slater's condition holds. Therefore, the existence of $\lambda^*$ is ensured, and the following inequality holds

$$f_k(z_k^*) = f_k(z_k^*) + \langle h_k(z_k^*), \lambda_k^* \rangle \leq f_k(\tilde{z}) + \langle h_k(\tilde{z}), \lambda_k^* \rangle \leq f_k(\tilde{z}) + \|\lambda_k^*\|_1 \max_i\{(h_k(\tilde{z}))_i\}.$$

Rearranging terms in the above inequality, we have

$$\|\lambda_k^*\|_1 \leq \frac{f_k(\tilde{z}) - f_k(z_k^*)}{-\max_i\{(h_k(\tilde{z}))_i\}}.$$

$\square$

We first provide the formal statement of the theorem and then provide the convergence.

**Theorem 5** (Formal Statement of Theorem 2)**.** *Suppose Assumption 1 hold. Consider Algorithm 3. Let $\sigma = 2\max\{L_f, L_c\}$, $\gamma_t = t + t_0 + 3$, $\eta_t = \frac{\rho_f(t+t_0+1)}{2}$, $\tau_t = \frac{4(L_g + 2\rho_h D_{\mathcal{Z}})^2}{\rho_f(t+1)}$, $\theta_t = \frac{t+t_0+2}{t+t_0+3}$, where $t_0 = \frac{\rho_f + B\rho_h}{\rho_f} + 1$, $B = B_0 + 1$ and $B_0$ defined in Lemma 7. Let $N \geq \log\left(\frac{\epsilon}{4(T+2)^2(L_g+\alpha)D_{\mathcal{Y}}D_{\mathcal{Z}}^2}\right) / \log(1 - \frac{\alpha}{L_g+2\alpha})$, $T \geq \mathcal{O}(\frac{1}{\sqrt{\epsilon}})$. We have*

$$f_k(\tilde{z}_{k+1}) - f_k(z_k^*) \leq \epsilon$$
$$\max_j\{(h_k(\tilde{z}_{k+1}))_j\} \leq \epsilon.$$

The proof is as follow.

We first define some notations that will be used later. By Proposition 2, we have both $f_k(z)$ and $h_k(z)$ are $\mu = \frac{\sigma}{2}$ strongly convex function. Let $\hat{d}_t = (1 + \theta_t)\hat{h}_k(z_t; \hat{y}_N^t) - \theta_t\hat{h}_k(z_{t-1}; \hat{y}_N^{t-1})$, $d_t = (1 + \theta_t)h(z_t) - \theta_t h(z_{t-1})$, and $\xi_t = \hat{h}_k(z_t; \hat{y}_N^t) - \hat{h}_k(z_{t-1}; \hat{y}_N^{t-1})$. Moreover, we specify $\mathcal{L}_k(z_t, \lambda_{t+1}; \hat{y}_N^t) = f_k(z_t) + \langle \lambda_{t+1}, \hat{h}_k(z_t; \hat{y}_N^t) \rangle$ and the gradient of Laguragian as $\widehat{\nabla}_z\mathcal{L}_k(z_t, \lambda_{t+1}; \hat{y}_N^t) = \nabla f_k(z_t) + \langle \lambda_{t+1}, \widehat{\nabla}h_k(z_t; \hat{y}_N^t) \rangle$. Further define the primal-dual gap function as

$$Q(w, \tilde{w}) := f_k(z) + \tilde{\lambda}h_k(z) - (f_k(\tilde{z}) + \lambda h_k(\tilde{z})),$$

where $w = (z, \lambda)$, $w = (\tilde{z}, \tilde{\lambda}) \in \mathcal{Z} \times \Lambda$ are primal-dual pairs.

Consider the update of $\lambda$ in eq. (13). Applying Lemma 5 with $G = -\hat{d}_t/\tau_t$, $\mathcal{Z} = \Lambda$, $\bar{z} = \lambda_{t+1}$, $z = \lambda_t$ and letting $\tilde{z} = \lambda$ be an arbitrary point inside $\Lambda$, we have

$$-(\lambda_{t+1} - \lambda)\hat{d}_t \leq \frac{\tau_t}{2}\left((\lambda - \lambda_t)^2 - (\lambda_{t+1} - \lambda_t)^2 - (\lambda - \lambda_{t+1})^2\right). \tag{44}$$

Similarly, consider the update of $z$ in eq. (14). Applying Lemma 5 with $G = \widehat{\nabla}_z\mathcal{L}_k(z_t, \lambda_{t+1}; \hat{y}_N^t)/\eta_t$, we obtain

$$\langle \widehat{\nabla}_z\mathcal{L}_k(z_t, \lambda_{t+1}; \hat{y}_N^t), z_{t+1} - z \rangle \leq \frac{\eta_t}{2}\left((z - z_t)^2 - (z_{t+1} - z_t)^2 - (z - z_{t+1})^2\right). \tag{45}$$

Recall that $f_k(z)$ and $h_k(z)$ are $L$-gradient Lipschitz. This implies

$$\langle \nabla f_k(z_t), z_{t+1} - z_t \rangle \geq f_k(z_{t+1}) - f_k(z_t) - \frac{L\|z_t - z_{t+1}\|_2^2}{2}, \tag{46}$$

$$\langle \nabla h_k(z_t), z_{t+1} - z_t \rangle \geq h_k(z_{t+1}) - h_k(z_t) - \frac{L\|z_t - z_{t+1}\|_2^2}{2}. \tag{47}$$

Recall that both $f_k$ and $h_k$ are $\mu$-strongly convex function. These two properties yield

$$\langle \nabla f_k(z_t), z_t - z \rangle \geq f_k(z_t) - f_k(z) + \frac{\mu\|z - z_t\|_2^2}{2}, \tag{48}$$

$$\langle \nabla h_k(z_t), z_t - z \rangle \geq h_k(z_t) - h_k(z) + \frac{\mu \|z - z_t\|_2^2}{2}. \tag{49}$$

Consider the exact gradient of Lagrangian with respect to the primal variable, we have

$$\langle \nabla_z \mathcal{L}_k(z_t, \lambda_{t+1}), z_{t+1} - z \rangle$$
$$= \langle \nabla f_k(z_t) + \lambda_{t+1} \nabla h_k(z_t), z_{t+1} - z_t \rangle$$
$$= \langle \nabla f_k(z_t), z_{t+1} - z \rangle + \langle \nabla f_k(z_t), z - z_t \rangle + \lambda_{t+1} \langle \nabla h_k(z_t), z_{t+1} - z \rangle + \lambda_{t+1} \langle \nabla h_k(z_t), z - z_t \rangle$$
$$\overset{(i)}{\geq} f_k(z_{t+1}) - f_k(z) + \lambda_{t+1}(h_k(z_{t+1}) - h_k(z)) - \frac{L(1 + \lambda_{t+1})\|z_{t+1} - z_t\|_2^2}{2} + \frac{\sigma(1 + \lambda_{t+1})\|z - z_t\|_2^2}{2}, \tag{50}$$

where $(i)$ follows from combining eqs. (46) to (49).

Combining eqs. (45) and (50) yields

$$f_k(z_{t+1}) - f_k(z) \leq \langle \nabla_z \mathcal{L}_k(z_t, \lambda_{t+1}) - \widehat{\nabla}_z \mathcal{L}_k(z_t, \lambda_{t+1}; \hat{y}_N^t), z_{t+1} - z \rangle + \lambda_{t+1}(h_k(z) - h_k(z_{t+1}))$$
$$+ \frac{\eta_t - \mu(1 + \lambda_{t+1})}{2} \|z - z_t\|_2^2 - \frac{\eta_t - L(1 + \lambda_{t+1})}{2} \|z_{t+1} - z_t\|_2^2$$
$$- \frac{\eta_t}{2} \|z - z_{t+1}\|_2^2. \tag{51}$$

Recall the definition of $\xi_t = \hat{h}_k(z_t; \hat{y}_N^t) - \hat{h}_k(z_{t-1}; \hat{y}_N^{t-1})$. Substituting it into eq. (44) yields

$$0 \leq -(\lambda - \lambda_{t+1})\hat{h}_k(z_t; \hat{y}_N^t) - (\lambda_{t+1} - \lambda)\xi_{t+1} + \theta_t(\lambda_{t+1} - \lambda)\xi_t$$
$$+ \frac{\tau_t}{2}\left((\lambda - \lambda_t)^2 - (\lambda_{t+1} - \lambda_t)^2 - (\lambda - \lambda_{t+1})^2\right). \tag{52}$$

Let $w = (z, \lambda)$ and $w_{t+1} = (z_{t+1}, \lambda_{t+1})$. By the definition of the primal-dual gap function, we have

$$Q(w_{t+1}, w)$$
$$= f_k(z_{t+1}) + \lambda h_k(z_{t+1}) - f_k(z) - \lambda_{t+1} h_k(z)$$
$$\overset{(i)}{\leq} \langle \nabla_z \mathcal{L}_k(z_t, \lambda_{t+1}) - \widehat{\nabla}_z \mathcal{L}_k(z_t, \lambda_{t+1}; \hat{y}_N^t), z_{t+1} - z \rangle + (\lambda - \lambda_{t+1})h_k(z_{t+1})$$
$$+ \frac{\eta_t - \mu(1 + \lambda_{t+1})}{2}\|z - z_t\|_2^2 - \frac{\eta_t - L(1 + \lambda_{t+1})}{2}\|z_{t+1} - z_t\|_2^2 - \frac{\eta_t}{2}\|z - z_{t+1}\|_2^2$$
$$\overset{(ii)}{\leq} \langle \nabla_z \mathcal{L}_k(z_t, \lambda_{t+1}) - \widehat{\nabla}_z \mathcal{L}_k(z_t, \lambda_{t+1}; \hat{y}_N^t), z_{t+1} - z \rangle + (\lambda - \lambda_{t+1})(h_k(z_{t+1}) - \hat{h}_k(z_{t+1}; \hat{y}_N^{t+1}))$$
$$- (\lambda_{t+1} - \lambda)\xi_{t+1} + \theta_t(\lambda_{t+1} - \lambda)\xi_t + \frac{\tau_t}{2}\left((\lambda - \lambda_t)^2 - (\lambda_{t+1} - \lambda_t)^2 - (\lambda - \lambda_{t+1})^2\right)$$
$$+ \frac{\eta_t - \mu}{2}\|z - z_t\|_2^2 - \frac{\eta_t - L(B + 1)}{2}\|z_{t+1} - z_t\|_2^2 - \frac{\eta_t}{2}\|z - z_{t+1}\|_2^2, \tag{53}$$

where $(i)$ follows from eq. (51) and $(ii)$ follows from eq. (52) and $0 \leq \lambda_{t+1} \leq B$.

Now we proceed with $|h_k(z_t) - \hat{h}_k(z_t; \hat{y}_N^t)|$.

$$|h_k(z_t) - \hat{h}_k(z_t; \hat{y}_N^t)| = |g(x_t, y_\alpha^*) - g(x_t, \hat{y}_N^t)| \overset{(i)}{\leq} 2L_g\|y_\alpha^* - \hat{y}_N^t\|_2 \overset{(ii)}{\leq} L_g D_{\mathcal{Z}}\left(1 - \frac{\alpha}{L_g + 2\alpha}\right)^N, \tag{54}$$

where $(i)$ follows from Assumption 1 and $(ii)$ follows from the following Lemma 4 and $\|\hat{y}_0^t - y_\alpha^*(x_t)\|_2 \leq D_{\mathcal{Z}}$.

The following inequality follows immediately from the above eq. (54).

$$(\lambda - \lambda_{t+1})(h_k(z_t) - \hat{h}_k(z_t; \hat{y}_N^t)) \leq |\lambda - \lambda_{t+1}||h_k(z_t) - \hat{h}_k(z_t; \hat{y}_N^t)| \leq L_g B D_{\mathcal{Z}}\left(1 - \frac{\alpha}{L_g + 2\alpha}\right)^N. \tag{55}$$

By the definitions of $\nabla_z \mathcal{L}_k(z_t, \lambda_{t+1})$ and $\widehat{\nabla}_z \mathcal{L}_k(z_t, \lambda_{t+1}; \hat{y}_N^t)$, we have

$$\|\nabla_z \mathcal{L}_k(z_t, \lambda_{t+1}) - \widehat{\nabla}_z \mathcal{L}_k(z_t, \lambda_{t+1}; \hat{y}_N^t)\|_2$$

$$= \left\| \nabla f_k(z_t) + \lambda_{t+1} \nabla h_k(z_t) - \left( \nabla f_k(z_t) + \lambda_{t+1} \widehat{\nabla} h_k(z_t; \hat{y}_N^t) \right) \right\|_2$$

$$= \lambda_{t+1} \left\| \nabla g(x_t, y_\alpha^*(x_t)) - \nabla g(x_t, \hat{y}_N^t) \right\|_2 \overset{(i)}{\leq} \lambda_{t+1} L_g \| y_\alpha^*(x_t) - \hat{y}_N^t \|_2$$

$$\overset{(ii)}{\leq} B L_g D_{\mathcal{Z}} \left( 1 - \tfrac{\alpha}{L_g + 2\alpha} \right)^N, \tag{56}$$

where $(i)$ follows from Assumption 1 and $(ii)$ follows from Lemma 4, and because $\lambda_{t+1} \leq B$ and $\| \hat{y}_0 - \tilde{y}^*(x_t) \|_2 \leq D_{\mathcal{Z}}$.

By Cauchy-Schwartz inequality and eq. (56), we have

$$\langle \nabla_z \mathcal{L}_k(z_t, \lambda_{t+1}) - \widehat{\nabla}_z \mathcal{L}_k(z_t, \lambda_{t+1}; \hat{y}_N^t), z_{t+1} - z \rangle$$

$$\leq \| \widehat{\nabla}_z \mathcal{L}_k(z_t, \lambda_{t+1}) - \widehat{\nabla}_z \mathcal{L}_k(z_t, \lambda_{t+1}; \hat{y}_N^t) \|_2 \| z_{t+1} - z \|_2 \leq B L_g D_{\mathcal{Z}}^2 \left( 1 - \tfrac{\alpha}{L_g + 2\alpha} \right)^N. \tag{57}$$

By the definition of $\xi_t$, we have

$$\theta_t(\lambda_{t+1} - \lambda_t)\xi_t = \theta_t(\lambda_{t+1} - \lambda_t)(\hat{h}_k(z_t; \hat{y}_N^t) - \hat{h}_k(z_{t-1}; \hat{y}_N^{t-1}))$$

$$= \theta_t(\lambda_{t+1} - \lambda_t)(\hat{h}_k(z_t; \hat{y}_N^t) - h_k(z_t) - \hat{h}_k(z_{t-1}; \hat{y}_N^{t-1}) + h_k(z_{t-1}) + h_k(z_t) - h_k(z_{t-1}))$$

$$\leq \theta_t |\lambda_{t+1} - \lambda_t| \left( |\hat{h}_k(z_t; \hat{y}_N^t) - h_k(z_t)| + |\hat{h}_k(z_{t-1}; \hat{y}_N^{t-1}) - h_k(z_{t-1})| + |h_k(z_t) - h_k(z_{t-1})| \right)$$

$$\overset{(i)}{\leq} |\lambda_{t+1} - \lambda_t| \left( 2 L_g D_{\mathcal{Z}} \left( 1 - \tfrac{\alpha}{L_g + 2\alpha} \right)^N + M \| z_t - z_{t-1} \|_2 \right)$$

$$\overset{(ii)}{\leq} 2 B L_g D_{\mathcal{Z}} \left( 1 - \tfrac{\alpha}{L_g + 2\alpha} \right)^N + M |\lambda_{t+1} - \lambda_t| \| z_t - z_{t-1} \|_2$$

$$\overset{(iii)}{\leq} 2 B L_g D_{\mathcal{Z}} \left( 1 - \tfrac{\alpha}{L_g + 2\alpha} \right)^N + \tfrac{\tau_t}{2} (\lambda_{t+1} - \lambda_t)^2 + \tfrac{M^2}{2\tau_t} \| z_t - z_{t-1} \|_2^2, \tag{58}$$

where $(i)$ follows from eq. (54), $\theta_t \leq 1$, and $h_k(z)$ is $M$ Lipschitz continuous, $(ii)$ follows from $0 \leq \lambda_t, \lambda_{t+1} \leq B$, and $(iii)$ follows from Young's inequality.

Substituting eqs. (55), (57) and (58) into eq. (53) yields

$$Q(w_{t+1}, w) \leq -(\lambda_{t+1} - \lambda)\xi_{t+1} + \theta_t(\lambda_t - \lambda)\xi_t + 4 L B D_{\mathcal{Z}}^2 \left( 1 - \tfrac{\alpha}{L_g + 2\alpha} \right)^N$$

$$+ \tfrac{\tau_t}{2} \left( (\lambda - \lambda_t)^2 - (\lambda - \lambda_{t+1})^2 \right) + \tfrac{\eta_t - \mu}{2} \| z - z_t \|_2^2 - \tfrac{\eta_t}{2} \| z - z_{t+1} \|_2^2$$

$$+ \tfrac{M^2}{2\tau_t} \| z_t - z_{t-1} \|_2^2 - \tfrac{\eta_t - L(1+B)}{2} \| z_{t+1} - z_t \|_2^2. \tag{59}$$

Recall that $\gamma_t, \theta_t, \eta_t$ and $\tau_t$ are set to satisfy $\gamma_{t+1}\theta_{t+1} = \gamma_t$, $\gamma_t \tau_t \geq \gamma_{t+1}\tau_{t+1}$ and

$$\gamma_t(L(1+B) - \eta_t) + \tfrac{\gamma_{t+1} M^2}{\tau_{t+1}} \leq 0.$$

Multiplying $\gamma_t$ on both sides of eq. (59) and telescoping from $t = 0, 1, \ldots T-1$ yield

$$\sum_{t=0}^{T-1} \gamma_t Q(w_{t+1}, w) \leq -\gamma_{T-1}(\lambda_T - \lambda)\xi_T + 4 L B D_{\mathcal{Z}}^2 \left( 1 - \tfrac{\alpha}{L_g + 2\alpha} \right)^N \sum_{t=0}^{T-1} \gamma_t$$

$$+ \tfrac{\gamma_0 \tau_0}{2} (\lambda - \lambda_0)^2 + \tfrac{\gamma_0(\eta_0 - \mu)}{2} \| z - z_0 \|_2^2$$

$$- \tfrac{\gamma_{T-1}(\eta_{T-1} - L(B+1))}{2} \| z - z_T \|_2^2.$$

Divide both sides of the above inequality by $\Gamma_T = \sum_{t=0}^{T-1} \gamma_t$. We obtain

$$\tfrac{1}{\Gamma_T} \sum_{t=0}^{T-1} \gamma_t Q(w_{t+1}, w) \leq -\tfrac{\gamma_{T-1}(\lambda_T - \lambda)\xi_T}{\Gamma_T} + 4 L B D_{\mathcal{Z}}^2 \left( 1 - \tfrac{\alpha}{L_g + 2\alpha} \right)^N$$

$$+ \frac{\gamma_0 \tau_0}{2\Gamma_T}(\lambda - \lambda_0)^2 + \frac{\gamma_0(\eta_0 - \rho_f)}{2\Gamma_T}\|z - z_0\|_2^2$$
$$- \frac{\gamma_{T-1}(\eta_{T-1} - 3(\rho_f + B\rho_h))}{2\Gamma_T}\|z - z_T\|_2^2. \tag{60}$$

Similarly to the steps in eq. (58), we have

$$|(\lambda_T - \lambda)\xi_T| \le |\lambda_T - \lambda| \left( 2L_g D_{\mathcal{Z}} \left(1 - \frac{\alpha}{L_g + 2\alpha}\right)^N + M\|z_T - z_{T-1}\|_2 \right)$$
$$\le 2L_g B D_{\mathcal{Z}} \left(1 - \frac{\alpha}{L_g + 2\alpha}\right)^N + MBD_{\mathcal{Z}}.$$

Define $\bar{w} := \frac{1}{\Gamma_T} \sum_{t=0}^{T-1} \gamma_t w_{t+1}$. Noting that $Q(\cdot, w)$ is a convex function and substituting the above inequality into eq. (60) yield

$$Q(\bar{w}, w) \le \frac{1}{\Gamma_T} \sum_{t=0}^{T-1} \gamma_t Q(w_{t+1}, w)$$
$$\le \frac{2L_g B D_{\mathcal{Z}}}{\Gamma_T} \left(1 - \frac{\alpha}{L_g + 2\alpha}\right)^N + \frac{(L_g + 2\rho_h D_{\mathcal{Z}})BD_{\mathcal{Z}}}{\Gamma_T}$$
$$+ (L_g D_{\mathcal{Z}} + 3L_g)BD_{\mathcal{Z}} \left(1 - \frac{\alpha}{L_g + 2\alpha}\right)^N + \frac{\gamma_0(\eta_0 - \rho_f)}{2\Gamma_T}\|z - z_0\|_2^2$$
$$+ \frac{\gamma_0 \tau_0}{2\Gamma_T}(\lambda - \lambda_0)^2 - \frac{\gamma_{T-1}(\eta_{T-1} - 3(\rho_f + B\rho_h))}{2\Gamma_T}\|z - z_T\|_2^2. \tag{61}$$

Let $w = (z_k^*, 0)$. Then, we have

$$Q(\tilde{w}_k, w) = f_k(\tilde{z}_{k+1}) - f_k(z_k^*) - \bar{\lambda}_T h_k(z_k^*) \overset{(i)}{\ge} f_k(\tilde{z}_{k+1}) - f_k(z_k^*),$$

where $(i)$ follows from the fact $h_k(z_k^*) \le 0$ and $\bar{\lambda}_T = \frac{1}{\Gamma_T} \sum_{t=0}^{T-1} \gamma_t \lambda_{t+1} \ge 0$.

Substituting the above inequality into eq. (61) yields

$$f_k(\tilde{z}_{k+1}) - f_k(z_k^*) \le \frac{2L_g B D_{\mathcal{Z}}}{\Gamma_T} \left(1 - \frac{\alpha}{L_g + 2\alpha}\right)^N + \frac{(L_g + 2\rho_h D_{\mathcal{Z}})BD_{\mathcal{Z}}}{\Gamma_T}$$
$$+ (L_g D_{\mathcal{Z}} + 3L_g)BD_{\mathcal{Z}} \left(1 - \frac{\alpha}{L_g + 2\alpha}\right)^N + \frac{\gamma_0(\eta_0 - \rho_f)\|z_k^* - z_0\|_2^2}{2\Gamma_T}. \tag{62}$$

Recall that $(z_k^*, \lambda_k^*)$ is a Nash equilibrium of $\mathcal{L}_k(z, \lambda)$, we have

$$\mathcal{L}_k(\tilde{z}_{k+1}, \lambda_k^*) \ge \mathcal{L}_k(z_k^*, \lambda_k^*) \overset{\text{by def.}}{\Longleftrightarrow} f_k(\tilde{z}_{k+1}) + \lambda_k^* h_k(\tilde{z}_{k+1}) - f_k(z_k^*) \ge 0 \tag{63}$$

Let $w = (z_k^*, (\lambda_k^* + 1)\mathbf{I}(h_k(\tilde{z}_{k+1})))$, where $\mathbf{I}(x) = 0$ if $x \le 0$ and $\mathbf{I}(x) = 1$ otherwise. If $h_k(\tilde{z}_{k+1}) \le 0$, the constraint is satisfied. If $h_k(\tilde{z}_{k+1}) > 0$, we have

$$Q(\tilde{w}_k, w) = f_k(\tilde{z}_{k+1}) + (\lambda_k^* + 1)h_k(\tilde{z}_{k+1}) - f_k(z_k^*) - \lambda_k^* h_k(z_k^*). \tag{64}$$

Recall that $(z_k^*, \lambda_k^*)$ satisfies the KKT condition of $(P_k)$, i.e. $\lambda_k^* h_k(z_k^*) = 0$. Equations (61), (63) and (64) together yield,

$$h_k(\tilde{z}_{k+1}) = Q(\tilde{w}_k, w) - (f_k(\tilde{z}_{k+1}) + \lambda_k^* h_k(\tilde{z}_{k+1}) - f_k(z_k^*)) \le Q(\tilde{w}_k, w)$$
$$\le \left( \frac{2L_g B D_{\mathcal{Z}}}{\Gamma_T} + (L_g D_{\mathcal{Z}} + 3L_g)BD_{\mathcal{Z}} \right) \left(1 - \frac{\alpha}{L_g + 2\alpha}\right)^N + \frac{(L_g + 2\rho_h D_{\mathcal{Z}})BD_{\mathcal{Z}}}{\Gamma_T}$$
$$+ \frac{\gamma_0 \tau_0}{2\Gamma_T}(\lambda_k^* + 1)^2 + \frac{\gamma_0(\eta_0 - \rho_f)}{2\Gamma_T}\|z_k^*\|_2^2. \tag{65}$$

We thus conclude, by taking $T = \mathcal{O}(\frac{1}{\sqrt{\epsilon}})$, $N = \mathcal{O}(\log(\frac{1}{\epsilon}))$, eqs. (62) and (65) complete the proof.

# G. Proof of Theorem 3

## G.1. Supporting Lemmas

**Lemma 8.** *Suppose Assumptions 1 and 3 hold, $\tilde{z}_{k+1}$ is $\frac{\beta}{2K}$-accurate solution of $\mathrm{P}_k$, and the input $\tilde{z}_1$ is strictly feasible with respect to $\mathrm{P}_k$ with margin $\frac{\beta}{2K}$. Let $\sigma = \{2L_f, L_c\}$. Then, we have*

$$\frac{1}{K}\sum_{k=1}^K \|\tilde{z}_{k+1} - z_k^*\|_2^2 \le \frac{4\Delta_f}{\sigma K} + \frac{2\beta}{\sigma K},$$

*with $\Delta_f = \max_{z, z' \in \mathcal{Z}} |f(z) - f(z')|$.*

*Proof.* For $\tilde{z}_{k+1}$ with $k \ge 1$, the $\frac{\beta}{2K}$-accuracy implies that

$$f_k(\tilde{z}_{k+1}) - f_k(z_k^*) \le \tfrac{\beta}{2K}, \tag{66}$$

$$h_k(\tilde{z}_{k+1}) \le \tfrac{\beta}{2K}. \tag{67}$$

Then, we have $h(\tilde{z}_{k+1}) = h_k(\tilde{z}_{k+1}) + \frac{k\beta}{K} - \frac{\sigma}{2}\|\tilde{z}_{k+1} - \tilde{z}_{k+1}\|_2^2 \le \frac{(2k+1)\beta}{2K}$, which immediately implies that $h_{k+1}(\tilde{z}_{k+1}) = h(\tilde{z}_{k+1}) - \frac{(k+1)\beta}{K} \le -\frac{\beta}{2K}$. Thus, given that $\tilde{z}_1$ is $\frac{\beta}{2K}$ strictly feasible of problem $\mathrm{P}_1$, we conclude that $\tilde{z}_k$ is $\frac{\beta}{2K}$ strictly feasible of problem $\mathrm{P}_k$ through induction.

Let $\mathcal{L}_k(z) = f_k(z) + (\lambda_k^*)^\top h_k(z) + \mathbb{1}_{\mathcal{Z}}(z)$, where $\mathbb{1}_{\mathcal{Z}}(z)$ is the indicator function. We have $\mathcal{L}_k(z)$ is a strongly convex function over $\mathbb{R}^{n+m+2}$. Given any $\zeta \in \mathcal{N}_{\mathcal{Z}}(z)$, we have $\nabla f_k(z) + \langle \nabla h_k(z), \lambda_k^* \rangle + \zeta \in \partial \mathcal{L}(z)$ for all $z \in \mathcal{Z}$. Clearly $z_k^* \in \arg\min_{z \in \mathcal{Z}} \mathcal{L}_k(z)$. The optimality gives us that $0 \in \partial \mathcal{L}_k(z_k^*)$. And, due to the strong convexity of $f_k(z)$ and $h_k(z)$ and $\lambda_k^* \ge 0$, $\mathcal{L}_k(z)$ is $\frac{\sigma}{2}$-strongly convex function. Thus, we have

$$
\begin{aligned}
\frac{\sigma}{4}\|\tilde{z}_k - z_k^*\|_2^2 &\stackrel{(i)}{\le} \mathcal{L}_k(\tilde{z}_k) - \mathcal{L}_k(z_k^*) \\
&= f_k(\tilde{z}_k) + (\lambda_k^*)^\top h_k(\tilde{z}_k) - \big(f_k(z_k^*) + (\lambda_k^*)^\top h_k(z_k^*)\big) \\
&\stackrel{(ii)}{\le} f_k(\tilde{z}_k) - f_k(z_k^*),
\end{aligned}
\tag{68}
$$

where $(i)$ follows from the strong convexity of $\mathcal{L}_k$ and $0 \in \partial \mathcal{L}_k(z_k^*)$, and $(ii)$ follows from the complementary slackness $(\lambda_k^*)^\top h_k(\tilde{z}_k) = 0$ and $\tilde{z}_k$ is feasible for $h_k(z)$.

Combining eqs. (66) and (68), we have

$$
\begin{aligned}
\frac{\sigma}{4}\|\tilde{z}_k - z_k^*\|_2^2 &\le f_k(\tilde{z}_k) - f_k(\tilde{z}_{k+1}) + \tfrac{\beta}{2K} \\
&\stackrel{(i)}{\le} f_k(\tilde{z}_k) - f(\tilde{z}_{k+1}) + \tfrac{\beta}{2K} \\
&\stackrel{(ii)}{=} f(\tilde{z}_k) - f(\tilde{z}_{k+1}) + \tfrac{\beta}{2K},
\end{aligned}
\tag{69}
$$

where $(i)$ follows from the fact that $f_k(\tilde{z}_{k+1}) = f(\tilde{z}_{k+1}) + \frac{\sigma}{2}\|\tilde{z}_{k+1} - \tilde{z}_k\|_2^2$, and $(ii)$ follows from $f_k(\tilde{z}_k) = f(\tilde{z}_k)$, $k \in \mathbb{N}$.

Telescoping eq. (69) and utilizing the definition of $\hat{k}$, we obtain

$$\mathbb{E}\left[\|\tilde{z}_{\hat{k}} - z_{\hat{k}}^*\|_2^2\right] = \frac{1}{K}\sum_{k=1}^K \|\tilde{z}_k - z_k^*\|_2^2 \le \frac{4}{\sigma K}\left(f(\tilde{z}_1) - f(\tilde{z}_{K+1}) + \frac{\beta}{2}\right) \stackrel{(i)}{\le} \frac{4\Delta_f}{\sigma K} + \frac{2\beta}{\sigma K} \tag{70}$$

where $(i)$ follows from the definition $\Delta_f = \max_{z, z'} |f(z) - f(z')|$. $\qquad\square$

## G.2. Proof of Theorem 3

We first provide the formal statement of Theorem 3 as follows.

**Theorem 6** (Formal Statement of Theorem 3). *Suppose Assumption 1 holds. Given $\tilde{z}_1$ that is $\frac{\beta}{2K}$ strictly feasible of* $(\mathrm{P}_1)$. *Let $\sigma = 2\max\{L_f, L_c\}$, where $L_c$ is determined in Proposition 2. Set $K \geq \frac{8(B+1)\Delta_f}{\epsilon}$, and $\beta = \min\{\frac{\epsilon}{4B}, 2\Delta_f\}$.. Then we have $\tilde{z}_{\hat{k}}$ is an $\epsilon$-KKT point of eq. (6) in expectation that takes the randomness over $\hat{k}$.*

Assumption 1 ensures that there exists $M_f$ and $M_h$, such that $\|\nabla f(z)\|_2 \leq M_f$ and $\|\nabla(h(z))_i\|_2 \leq M_h$, thus we have $\|\nabla f_k(z)\|_2 \leq M_f + \sigma D_{\mathcal{Z}}$ and $\|\nabla(h_k(z))_i\|_2 \leq M_h + \sigma D_{\mathcal{Z}}$, with $D_{\mathcal{Z}} = \max_{z,z' \in \mathcal{Z}} \|z - z'\|_2$. Let $M = \max\{M_f, M_h\} + \sigma D_{\mathcal{Z}}$, where $M_f = \max_{z \in \mathcal{Z}}\{\|\nabla f(z)\|_2\}$ and $M_h = \max_{z \in \mathcal{Z}, i \in [q]}\{\|\nabla(h(z))_i\|_2\}$.

By the requirement of the algorithm, we have, for each $k = 1, \ldots, K$

$$f_k(\tilde{z}_k) - f_k(z_k^*) \leq \frac{\beta}{2K},$$
$$\max\{(h_k(\tilde{z}_k))_i\} \leq \frac{\beta}{2K}.$$

Applying Lemma 8, we have

$$\frac{1}{K}\sum_{k=1}^{K} \|\tilde{z}_k - z_k^*\|_2^2 \leq \frac{4\Delta_f}{\sigma K} + \frac{2\beta}{\sigma K}, \tag{71}$$

Moreover, the optimality of $(z_k^*, \lambda_k^*)$ for subproblem $\mathrm{P}_k$ shows that, there exists $\zeta_k \in \mathcal{N}_{\mathcal{Z}}(z_k^*)$ such that

$$\nabla f_k(z_k^*) + \langle \nabla h_k(z), \lambda_k^*\rangle + \zeta_k = 0. \tag{72}$$

Using the facts, $\nabla f_k(z_k^*) = \nabla f(z_k^*) + \sigma(z_k^* - \tilde{z}_k)$ and $\nabla h_k(z_k^*) = \nabla h(z_k^*) + \sigma \mathbf{1}(z_k^* - \tilde{z}_k)^\top$, eq. (72) implies

$$\nabla f(z_k^*) + \langle \nabla h(z_k^*), \lambda_k^*\rangle + \zeta_k = -(\|\lambda_k^*\|_1 + 1)\sigma(\tilde{z}_k - z_k^*).$$

Taking $\ell_2$-norm on both sides of the above equality, and using the upper bound of $\|\lambda_k^*\|_2$ in Assumption 3, we have

$$\|\nabla f(z_k^*) + \langle \lambda_k^* \nabla h(z_k^*)\rangle + \zeta_k\|_2 \leq (B+1)\sigma\|\tilde{z}_k - z_k^*\|_2. \tag{73}$$

Telescoping eq. (73) and applying eq. (71), we have

$$\mathbb{E}\left[\left\|\nabla f(z_{\hat{k}}^*) + \langle \lambda_{\hat{k}}^* \nabla h(z_{\hat{k}}^*)\rangle + \zeta_{\hat{k}}\right\|_2\right] \leq \frac{(B+1)(4\Delta_f + 2\beta)}{K},$$

Using the fact that $\zeta_k \in \mathcal{N}_{\mathcal{Z}}(z_k^*)$, we have

$$\mathbb{E}\left[\mathrm{dist}\left(\nabla f(z_{\hat{k}}^*) + \langle \lambda_{\hat{k}}^* \nabla h(z_{\hat{k}}^*)\rangle, -\mathcal{N}_{\mathcal{Z}}(z_{\hat{k}}^*)\right)\right] \leq \frac{(B+1)(4\Delta_f + 2\beta)}{K}. \tag{74}$$

Moreover we have

$$\sum_{i=1}^{q} |(\lambda_k^*)_i(h(z_k^*))_i| = \sum_{i=1}^{q}\left|(\lambda_k^*)_i\left((h_k(z_k^*))_i - \frac{\sigma}{2}\|\tilde{z}_k - z_k^*\|_2^2 + \frac{k\beta}{K}\right)\right| \stackrel{(i)}{\leq} \frac{B\sigma}{2}\|\tilde{z}_k - z_k^*\|_2^2 + \frac{kB\beta}{K},$$

where $q$ is the dimension of the constraint $h$, $(i)$ follows from the complementary slackness of $z_k^*$. Telescoping the above inequality, we obtain

$$\mathbb{E}\left[\sum_{i=1}^{q}\left|\left(\lambda_{\hat{k}}^*\right)_i\left(h(z_{\hat{k}}^*)\right)_i\right|\right] = \frac{1}{K}\sum_{k=1}^{K}\sum_{i=1}^{q}|(\lambda_k^*)_i(h(z_k^*))_i| \leq \frac{B(4\Delta_f + 2\beta)}{K} + \frac{K(K+1)B}{K^2}\cdot\beta. \tag{75}$$

Recall that $\mathbb{E}[h(z_{\hat{k}}^*)] = \frac{1}{K}\sum_{k=1}^{K}h(z_k^*) \leq \frac{(K+1)\beta}{K} \leq 2\beta$. Using the facts, $K \geq \frac{8(B+1)\Delta_f}{\epsilon}$, $\beta = \min\{\frac{\epsilon}{4B}, 2\Delta_f\}$, eq. (60) induces $\mathbb{E}\left[\|\tilde{z}_{\hat{k}} - z_{\hat{k}}^*\|_2^2\right] \leq \epsilon$, eqs. (74) and (75) imply that

$$\mathbb{E}\left[\mathrm{dist}\left(\nabla f(z_{\hat{k}}^*) + \langle \lambda_{\hat{k}}^* \nabla h(z_{\hat{k}}^*)\rangle, -\mathcal{N}_{\mathcal{Z}}(z_{\hat{k}})\right)\right] \leq \epsilon, \mathbb{E}\left[\sum_{i=1}^{q}\left|(\lambda_{\hat{k}}^*)_i(h(z_{\hat{k}}^*))_i\right|\right] \leq \epsilon.$$

# H. Gradients of the Relaxed Problem in Illustrative Example

The KKT reformulation of the problem in eq. (8) is

$$\min_{x,y,w,v\in\mathbb{R}} \quad -xy$$
$$\text{s.t.} \quad x^2 + y^2 - 1 - \xi \leq 0$$
$$g(x,y) - \xi \leq 0$$
$$x + 2wy + vG(x,y) = 0$$
$$w(x^2 + y^2 - 1 - \xi) = 0$$
$$v(g(x,y) - \xi) = 0,$$

where $G(x,y) := \nabla_y g(x,y)$ and it equals

$$G(x,y) = \begin{cases} 3(y - |x|)^2 & y \geq |x| \\ 0 & -|x| \leq y \leq |x| \\ -3(y + |x|)^2 & y \leq -|x| \end{cases}.$$

The final relaxed problem is

$$\min_{x,y,w,v\in\mathbb{R}} \quad -xy$$
$$\text{s.t.} \quad x^2 + y^2 - 1 - \xi \leq 0$$
$$g(x,y) - \xi \leq 0$$
$$x + 2wy + vG(x,y) - \beta \leq 0$$
$$-x - 2wy - vG(x,y) - \beta \leq 0$$
$$w(x^2 + y^2 - 1 - \xi) - \beta \leq 0$$
$$-w(x^2 + y^2 - 1 - \xi) - \beta \leq 0$$
$$v(g(x,y) - \xi) - \beta \leq 0$$
$$-v(g(x,y) - \xi) - \beta \leq 0.$$

Denote $h(z)$ as

$$h(z) = \begin{pmatrix} x^2 + y^2 - 1 - \xi \\ g(x,y) - \xi \\ x + 2wy + vG(x,y) - \beta \\ -x - 2wy - vG(x,y) - \beta \\ w(x^2 + y^2 - 1 - \xi) - \beta \\ -w(x^2 + y^2 - 1 - \xi) - \beta \\ v(g(x,y) - \xi) - \beta \\ -v(g(x,y) - \xi) - \beta \end{pmatrix}.$$

$\nabla f(z) = [-y; -x; 0; 0]$, and for the constrained function $h(z)$, we have

$$\nabla h(z) = \begin{pmatrix} 2x & 2y & 0 & 0 \\ \frac{\partial g(x,y)}{\partial x} & \frac{\partial g(x,y)}{\partial y} & 0 & 0 \\ 1 + v\frac{\partial G(x,y)}{\partial x} & 2w + v\frac{\partial G(x,y)}{\partial y} & 2y & G(x,y) \\ -1 - v\frac{\partial G(x,y)}{\partial x} & -2w - v\frac{\partial G(x,y)}{\partial y} & -2y & -G(x,y) \\ 2wx & 2wy & x^2 + y^2 - 1 - \xi & 0 \\ -2wx & -2wy & -(x^2 + y^2 - 1 - \xi) & 0 \\ v\frac{\partial g(x,y)}{\partial x} & v\frac{\partial g(x,y)}{\partial y} & 0 & g(x,y) - \xi \\ -v\frac{\partial g(x,y)}{\partial x} & -v\frac{\partial g(x,y)}{\partial y} & 0 & -g(x,y) + \xi \end{pmatrix},$$

$$(76)$$

where the gradient of the $i$th entry equals to the $i$th row of the above matrix and we have

$$g(x, y) := \begin{cases} (y - |x|)^3, & y \geq |x| \\ 0, & -|x| \leq y \leq |x| \\ -(y + |x|)^3, & y \leq -|x| \end{cases}.$$

$$\frac{\partial g(x,y)}{\partial x} = \begin{cases} -3\text{sgn}(x)(|x| - y)^2 & y \geq |x| \\ 0 & |y| \leq |x| \\ -3\text{sgn}(x)(|x| + y)^2 & y \leq -|x| \end{cases},$$

with $\text{sgn}(x) = 1$ if $x \geq 0$ and $\text{sgn}(x) = -1$ otherwise.

$$\frac{\partial g(x,y)}{\partial y} = G(x, y) = \begin{cases} 3(y - |x|)^2 & y \geq |x| \\ 0 & |y| \leq |x| \\ -3(y + |x|)^2 & y \leq -|x| \end{cases},$$

$$\frac{\partial G(x,y)}{\partial x} = \begin{cases} 6\text{sgn}(x)(|x| - y) & y \geq |x| \\ 0 & |y| \leq |x| \\ -6\text{sgn}(x)(y + |x|) & y \leq -|x| \end{cases},$$

$$\frac{\partial G(x,y)}{\partial y} = \begin{cases} 6(y - |x|) & y \geq |x| \\ 0 & |y| \leq |x| \\ -6(y + |x|) & y \leq -|x| \end{cases}.$$

