# OpenReview forum: "AdaProx: A Novel Method for Bilevel Optimization under Pessimistic Framework"
_CPAL.cc/2025/Proceedings_Track — CPAL 2025 (Proceedings Track) Poster_

### Official Review · Reviewer_y421 · 2025-01-06
**Review of "AdaProx: A Novel Method for Bilevel Optimization under Pessimistic Framework"**

**Rating:** 7
**Confidence:** 3

**Review:**

## Summary

This paper introduces **AdaProx**, a novel algorithmic framework for *pessimistic bilevel optimization* (PBO) problems. Unlike optimistic bilevel problems, PBO involves an outer-level minimization over *worst-case* values of the inner-level minima, thereby creating a more challenging minimax structure. To tackle this, the authors transform the PBO into a single-level constrained optimization problem using *KKT conditions*, then introduce two core ideas: **(i)** adaptive relaxation, which alleviates potential infeasibility in constraints, and **(ii)** accuracy-level tuning, which addresses bias in gradient estimation. The paper provides **theoretical guarantees**, showing that AdaProx converges to an $\varepsilon$-KKT point in sublinear time under certain assumptions, along with a characterization of its computational complexity. Experimental results on illustrative and robust hyper-representation learning tasks validate the approach, indicating consistency with theoretical predictions.

---

## Evaluation
Overall, this paper makes a noteworthy contribution by proposing a *first-order, provably convergent* algorithm for *nonlinear* PBO, a domain that has been underexplored in bilevel optimization research. The **theoretical analysis** is thorough and carefully addresses convergence, complexity, and issues related to constraint handling. Nonetheless, the **experimental scope** is limited to synthetic scenarios and smaller-scale demonstrations, leaving questions about real-world scalability and empirical robustness.

### Pros

1. **Strong Theoretical Underpinnings**
   - The paper rigorously derives convergence guarantees for a nonlinear PBO setting, providing new insights into gradient-based bilevel algorithms.

2. **Algorithmic Innovations**
   - ***Adaptive Relaxation***: Dynamically adjusts constraint feasibility, addressing an important practical hurdle in constrained bilevel problems.
   - ***Accuracy-Level Techniques***: Mitigates bias in gradient approximations, contributing to a more reliable solution process.

3. **Relevant Applications**
   - The focus on *worst-case* or robust scenarios (e.g., robust hyperparameter tuning, adversarial learning) is timely and aligns with current trends in robust machine learning.

4. **Comprehensive Reformulation**
   - The single-level approach grounded in **KKT conditions** is clearly explained, demonstrating a well-structured transition from bilevel to constrained optimization.


### Cons

1. **Limited Experimental Scope**
   - Although the paper’s experiments confirm theoretical insights, they are largely illustrative or synthetic. No large-scale or real-world benchmarks are provided, curtailing evidence of practical impact.

2. **No Baseline Comparisons**
   - While principled algorithms for nonlinear PBO may be scarce, researchers might still tackle these problems using heuristic or approximate methods (e.g., naive bilevel solvers, multi-start gradient-based methods, or modified variants of optimistic bilevel algorithms). The paper would benefit from comparing AdaProx against these plausible baselines, even if they are not rigorously designed for PBO, to demonstrate its practical advantages.

3. **Scalability Concerns**
   - The computational complexity is analyzed theoretically, but practical scalability—especially in high-dimensional settings—remains largely untested.

4. **Dense Technical Exposition**
   - While methodically presented, sections on theoretical proofs and algorithmic details might benefit from including more intuitive explanations or practical takeaways.

---

### Official Review · Reviewer_xBg5 · 2025-01-06

**Rating:** 8
**Confidence:** 3

**Review:**

**Summary**: The authors present challenges of relaxations in PBO and propose a new reformulation to standard constraint convex optimization with a first-order method backed by theoretical support.

**Strengths**:
- The paper is clearly written and easy to follow.
- Theoretical convergence results.
- The adaptive constraint relaxation.
- Considering Learning Robust Hyper-representation experiment

**Questions**:
- How is the optimization problem in eq.8 a bi-level optimization problem? The constraint is not an optimization problem.

---

### Official Review · Reviewer_1oUq · 2025-01-12

**Rating:** 5
**Confidence:** 4

**Review:**

This paper tries to address pessimistic bilevel optimization, a less explored class of optimization problems relevant to robust machine learning applications such as hyperparameter learning and adversarial robustness. Unlike optimistic bilevel optimization, PBO maximizes the outer objective over the worst-case solutions of the inner problem, making it more complex. Existing methods are either limited to linear/quadratic settings or lack convergence guarantees for general nonlinear PBO problems. The authors provide rigorous theoretical analysis, demonstrating that AdaProx achieves sublinear convergence to an $\epsilon$-KKT point. Empirical results on synthetic examples and a robust hyper-representation learning task validate the effectiveness of the approach, with AdaProx-SG showing superior constraint handling at the cost of stability compared to AdaProx-PD. This work represents a significant advancement by providing the first provably convergent algorithm for nonlinear PBO problems.

Strengths:
1. The paper pioneers a principled approach to nonlinear PBO by reformulating it as a constrained optimization problem.
2. This paper provides detailed theoretical guarantees for convergence, addressing practical issues like gradient bias and non-convex constraints.

Weaknesses:
1. While the paper tackles a challenging problem, it assumes strong smoothness conditions, which may limit real-world applicability. For example, in the real-world scenarios like hyper-parameter tuning or adversarial robustness.
2. The experiments focus on synthetic and small-scale datasets. Expanding to real-world tasks could strengthen the empirical claims. Some existing work on bi-level optimization using large-scale datasets:

    > Revisiting and Advancing Fast Adversarial Training Through The Lens of Bi-Level Optimization, ICML'22

3. Some mathematical notations and definitions (e.g., constraint relaxations and dual updates) could be better explained for accessibility.

---

### Meta-Review · Area_Chair_ZaL5 · 2025-02-05

**Recommendation:** Accept (Poster)
**Confidence:** 5

**Metareview:**

The paper proposes AdaProx, a novel method for pessimistic bilevel optimization (PBO), which involves maximizing the outer objective over the worst-case solutions of the inner problem. The approach transforms the PBO into a constrained optimization problem using KKT conditions and introduces adaptive strategies for constraint relaxation and accuracy-level tuning to manage gradient estimation biases.

All reviewers agree on the contributions of the paper. Despite some limitations regarding assumptions and empirical testing scope, the paper sets a solid foundation for future research on PBO.

---

### Decision · Program_Chairs · 2025-02-11

Accept (Poster)